# Efficient circular RNA synthesis for potent rolling circle translation

Yifei Du [1] ✉, Philipp Konrad Zuber[1], Huajuan Xiao[2], Xueyan Li[1],
Yuliya Gordiyenko[1] & V. Ramakrishnan [1] ✉

Circular RNA (circRNA) is a candidate for next-generation messenger RNA therapeutics owing to its remarkable stability. Here we describe *trans*-splicing-based methods for the synthesis of circRNAs over 8,000 nucleotides. The methods are independent of bacterial sequences, outperform the permuted intron–exon method and allow for the incorporation of RNA modifications. The resulting unmodified circRNAs, which incorporate sequences from human 28S ribosomal RNA, display low immunogenicity and are translated more efficiently than permuted intron–exon-derived circRNAs. Additionally, by using viral internal ribosomal entry sites for rolling circle translation, we show that ribosomes can efficiently read through highly structured internal ribosomal entry sites, enhancing the efficiency of rolling circle translation by over 7,000-fold with respect to previous constructs. The efficient and reliable production of circRNA may facilitate its therapeutic use.

When exogenous messenger RNA is introduced into cells, a gene of interest (GOI) can be expressed, forming the basis of mRNA therapeutics. Despite promising results in the 1990s, the development of mRNA therapeutics progressed slowly due to challenges such as instability, high immunogenicity and inefficient delivery[1]. Recent advances in nucleotide modification and RNA delivery technologies have led to the success of mRNA vaccines[2]. However, mRNA instability remains a challenge for other applications such as protein replacement therapy[2].

Owing to their exonuclease resistance, circular RNAs (circRNAs) are more stable than mRNAs and are thus promising alternatives to mRNAs for therapeutics[2,3]. Natural circRNAs can serve as sponges for microRNAs and proteins but are also templates for translation[4]. In the presence of an internal ribosome entry site (IRES) for initiation, circRNAs can be translated efficiently[5–8]. CircRNAs that lack an in-frame termination codon allow ribosomes to translate completely around the circRNA multiple times, resulting in a polyprotein, a process known as rolling circle translation (RCT)[9,10]. The polyprotein can in principle be cleaved by a protease or a self-cleaving sequence, thus allowing multiple copies of the GOI to be made in a single round of initiation. In principle, RCT can be 100-fold more efficient than single-shot translation[9,10]. However, RCT encounters two challenges: low initiation

efficiency and accessory sequences introduced by currently used in vitro circularization methods[6,11].

CircRNAs can be synthesised in vitro by either chemical or enzymatic ligation or the group I intron-based permuted intron–exon (PIE) method[12]. Group I introns are well characterized *cis* ribozymes that join flanking exons by a two-step transesterification reaction[13] (Fig. 1a). The PIE relocates the exon 1 and 5′ half of the intron to 3′ of the exon 2, thereby connects the flanking exons[14,15]. With this arrangement, PIE can generate circRNAs efficiently[6,11] (Fig. 1b). However, the PIE has limitations, including low yields for long circRNAs, immunogenicity resulting from bacterial sequences and its inability to synthesize modified circRNAs[6,8,16].

The identification of circRNAs poses another challenge. Native and formaldehyde agarose gels cannot separate circRNAs from their linear counterparts[4,17]. Urea–polyacrylamide gels effectively distinguish circRNAs from their linear forms but are limited to <500 nt (refs. [4,18]). The E-gel system can separate circRNAs and their linear counterparts but exhibits batch-to-batch variation[6,19]. Therefore, a generally applicable, reliable gel system is desirable for the identification of circRNAs.

Here, we report highly efficient and precise circRNA synthesis techniques, along with simple analytical methods for the characterization

[1]MRC Laboratory of Molecular Biology, Cambridge, UK. [2]Independent researcher, Cambridge, UK. ✉e-mail: yifei.du18@gmail.com;
ramak@mrc-lmb.cam.ac.uk

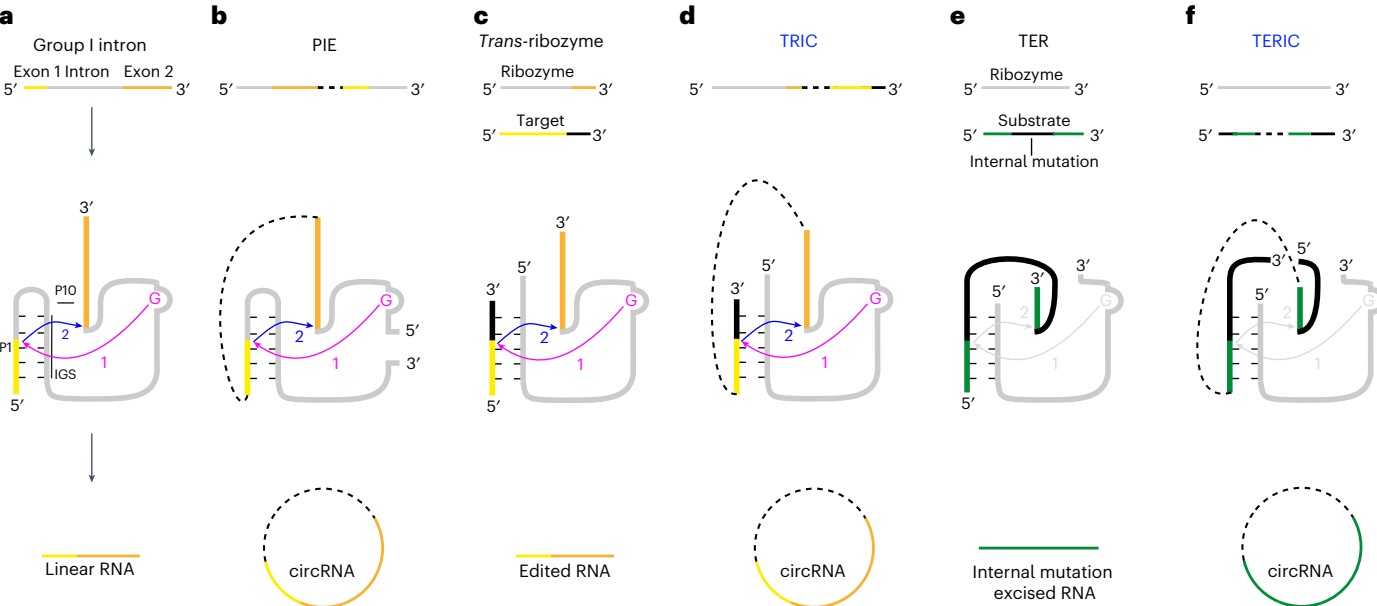

**Fig. 1 | Conceptualization of the TRIC. a**, Group I introns form P1 and P10 interactions between their IGS and the flanking exons, allowing joining the exons through a two-step transesterification reaction initiated by GTP (magenta). **b**, The PIE method for RNA circularization. **c**, Group I introns can act as a *trans*-ribozyme to edit a target sequence. **d**, The TRIC connects the target sequence

to 3′ end of the *trans*-ribozyme, thereby producing circRNAs. **e**, Group I introns function as *trans*-excision ribozymes (TER) to remove a middle (bridge) sequence from their target sequences. **f**, The *trans*-excision ribozyme-based circularization (TERIC) produces circRNAs.

of circRNAs. These methods have potential benefits for therapeutic applications of circRNAs.

## Results

### The TRIC synthesizes circRNAs efficiently

Group I introns can also act as *trans*-ribozymes[20,21] (Fig. 1c). Therefore, once the target sequence is connected to the 3′ end of the ribozyme (Fig. 1d), the splicing reaction produces circRNAs. This *trans*-ribozyme-based circularization (TRIC) approach keeps the intact ribozyme upstream of target sequences and thus allows efficient local folding of the entire ribozyme.

To test the TRIC method, we selected the Anabaena transfer (t) RNA[Leu] group I intron as it is short (249 nt) and highly active (Extended Data Fig. 1)[6,22]. The Anabaena intron is situated in the tRNA anticodon arm (ACA) and utilizes its internal guide sequence (IGS) to form P1 and P10 interactions with flanking exons[13] (Fig. 2a and Extended Data Fig. 1). To retain P1 and P10, part of the tRNA[Leu] was preserved in TRIC-V0 for circularizing a 141 nt 3×Flag sequence (Fig. 2b). We analysed the in vitro transcription (IVT) sample using a 12% urea–polyacrylamide gel and identified the 403 nt full-length precursor (FL), the 257 nt spliced linear intron, and two unknown species (I and II; Fig. 2c, left). In a subsequent 6% urea–polyacrylamide gel, the electroeluted RNAs I and II both exhibited a minor species that either matched the linear intron or the nicked 3×Flag, suggesting that I is the circular intron and II is the circular 3×Flag (circ3×Flag; Fig. 2c, right). The circularity and identity were further confirmed by reverse transcription followed by PCR (RT–PCR) and Sanger sequencing (Fig. 2d,e). Due to the apparent presence of circ3×Flag, we concluded that the TRIC-V0 construct circularizes the 3×Flag sequence efficiently during IVT (Extended Data Fig. 2a,b).

An extended guide sequence (EGS) and the internal loop between EGS and IGS are key for enhancing the efficiency of *trans*-ribozymes[23,24]. Therefore, we next constructed TRIC-V1 by introducing a 20 or 23 bp EGS and internal loops to TRIC-V0 (Fig. 2f, right and Extended Data Fig. 2d). All tested V1 variants circularized the 3×Flag efficiently (Fig. 2f, left). Since V1.0 yielded the highest circRNA to intron ratio, we chose this construct for subsequent optimizations (Extended Data Fig. 2a,c).

### A new protocol enables synthesis of circRNAs >8,000 nt

Next, we wanted to evaluate the performance of V1.0 in circularizing long GOIs and compare it with the PIE (Fig. 2g). Gel analysis suggested that V1.0 has a higher cotranscriptional splicing efficiency than the PIE for the Coxsackievirus B3 (CVB3)-contained enhanced green fluorescent protein (EGFP), Firefly luciferase (Fluc), spike protein of SARS-CoV-2 (Spike) and Cas9 (Fig. 2h). Twenty minutes of post-transcriptional circularization at 55 °C reduced the amount of FL for all constructs. However, due to severe nicking and degradation, we were unable to identify circRNA bands.

$Mg^{2+}$ ions and elevated temperatures are the primary causes of RNA nicking[6]. Since both are essential for IVT and circularization, we reasoned that suppressing cotranscriptional circularization and accelerating post-transcriptional circularization are rational ways to reduce nicking of the circRNA product. We titrated $Mg^{2+}$ in the IVT reactions and found that cotranscriptional circularization was suppressed using ≤16 mM $Mg^{2+}$ at 24 mM NTP concentration[25] (Extended Data Fig. 2e). Notably, IVT yield is only significantly reduced when the $Mg^{2+}$ concentration is lower than 12 mM (Extended Data Fig. 2f). Using the generated FL for post-transcriptional circularization, V1.0 circularized all GOIs efficiently, including the 8,706 nt Factor 8 (Fig. 2i and Extended Data Fig. 2g–j) (see below for circRNA band assignment). Prolonged circularization decreased the FL, but also increased circRNA nicking. Moreover, a direct comparison between V1.0 and PIE at 20 min of circularization revealed comparable efficiencies (ratio of converted FL) (Extended Data Fig. 2k). Notably, the absence of circular Spike, Cas9 and Factor 8 in the widely used co-post-transcriptional circularization protocol (Fig. 2h) suggests that the FL post-transcriptional circularization protocol is essential for synthesis of circRNAs >5,000 nt.

### Native agarose gels can separate circRNAs from their linear counterparts

It is assumed that native agarose gels cannot resolve circRNA and its linear form[4,11]. However, we consistently observed two distinct bands, which were both resistant to RNase R exonuclease treatment, at the

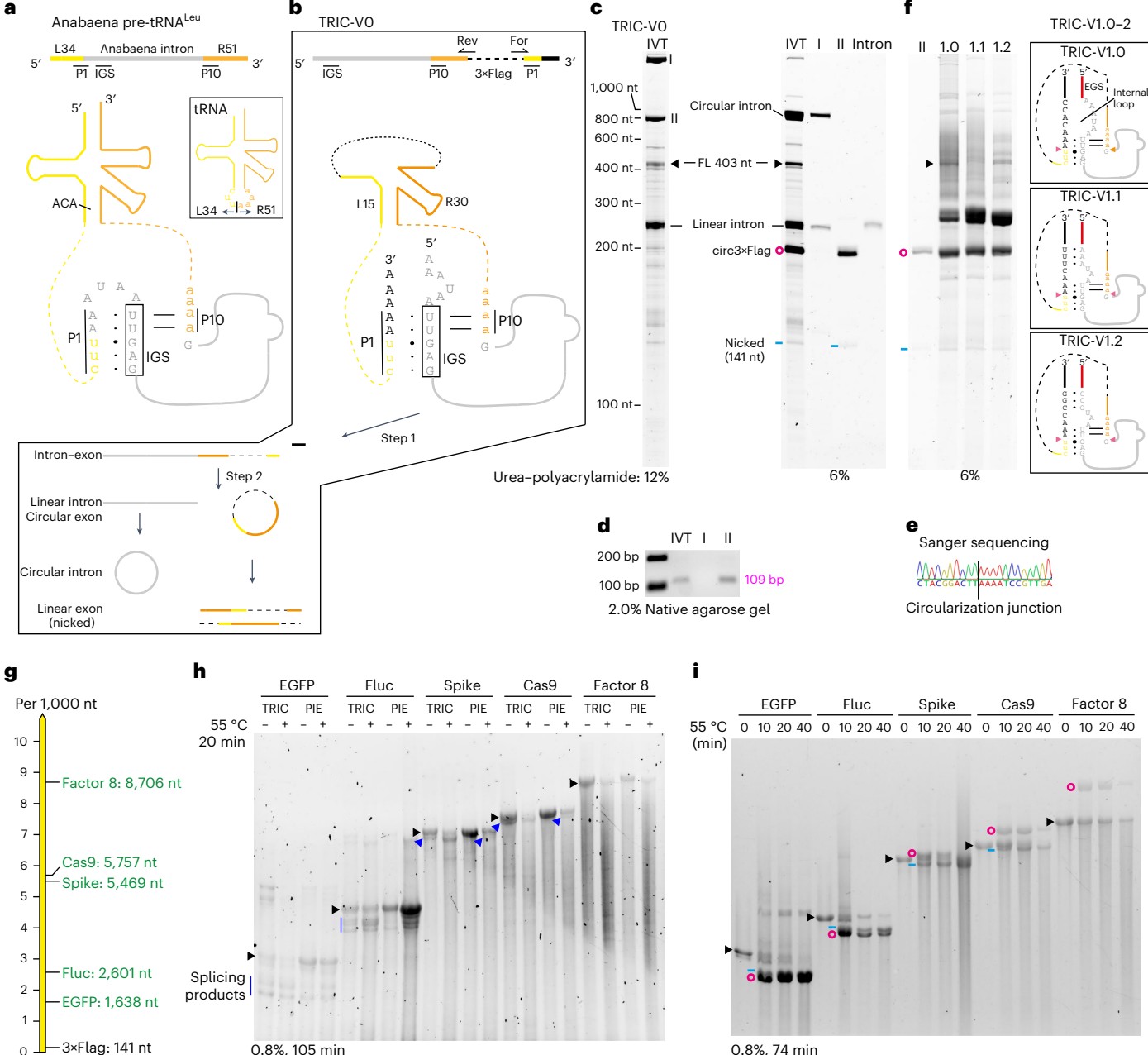

**Fig. 2 | TRIC synthesize circRNAs efficiently. a**, A scheme of the Anabaena tRNA^Leu group I intron. The Anabaena intron divides the tRNA^Leu into a 34 nt left half (L34) and a 51 nt right half (R51). **b**, TRIC-V0 retains L15 and R30 of the tRNA sequence. Insert: the two-step reaction of the V0. **c**, Left: cotranscriptionally circularized sample of V0-3×Flag analysed on a 12% urea–polyacrylamide gel. Right: cotranscriptionally circularized sample of V0-3×Flag and electroeluted linear intron, and band I and II loaded on a 6% urea–polyacrylamide gel. The RNA ladder is labelled on the left. The black triangle, blue line and red circle indicate the FL, nicked circRNA and circRNA, respectively. This applies to all the following figures. **d**, RT–PCR using cotranscriptionally circularized sample of V0-3×Flag, and RNAs band I and II as templates. The positions of forward (For) and reverse (Rev) primers are indicated in **b**. **e**, Sanger sequencing of the RT–PCR product from band II RNA. **f**, Cotranscriptionally circularized samples of V1 variants

analysed by 6% urea–polyacrylamide gel. The electroeluted band II was used as a reference. Inserts: diagrams of V1.0–2 showing introduction of the EGS and the internal loop. **g**, Five GOIs spanning from 1,638 nt (EGFP) to 8,706 nt (Factor 8) were assembled into V1.0 and PIE constructs. All resulting circRNAs contained a CVB3 IRES and a CDS of EGFP, Fluc, Spike-EGFP, Cas9-EGFP or Factor 8-EGFP. For each GOI, circRNAs generated by the V1.0 and the PIE have identical sequence. **h**, Cotranscriptionally circularized samples of V1.0 and PIE for long GOIs were analysed on a 0.8% native agarose gel operated at 25 W constant power. Cotranscriptional splicing products are indicated with vertical blue lines or blue arrows. Additional post-transcriptional circularization was achieved by adding 2 mM GTP and heating at 55 °C for 20 min. **i**, FL samples of V1.0 for long GOIs were circularized for various durations (0, 10, 20 or 40 min) and then analysed in a 0.8% native agarose gel.

rough position of circEGFP in a 0.8% agarose gel (Fig. 3a,b). When the gel-purified lower band was linearized by RNase H, it migrated similarly to the upper band, indicating that the lower and the upper bands are circular and nicked EGFP, respectively (Fig. 3c). This suggests that circEGFP moves faster than nicked EGFP in native agarose gels.

To see whether this is a general trend, we analysed circRNAs spanning lengths from 813 to 5,757 nt on 0.8%, 1.5% and 3% native agarose gels (Fig. 3d–f). In the 0.8% gel, the P2A-EGFP (no CVB3, 813 nt) and the Fluc (2,601 nt) behaved similarly to the EGFP (Fig. 3d). For the Spike, two largely overlapping bands appeared below the FL. However, a new

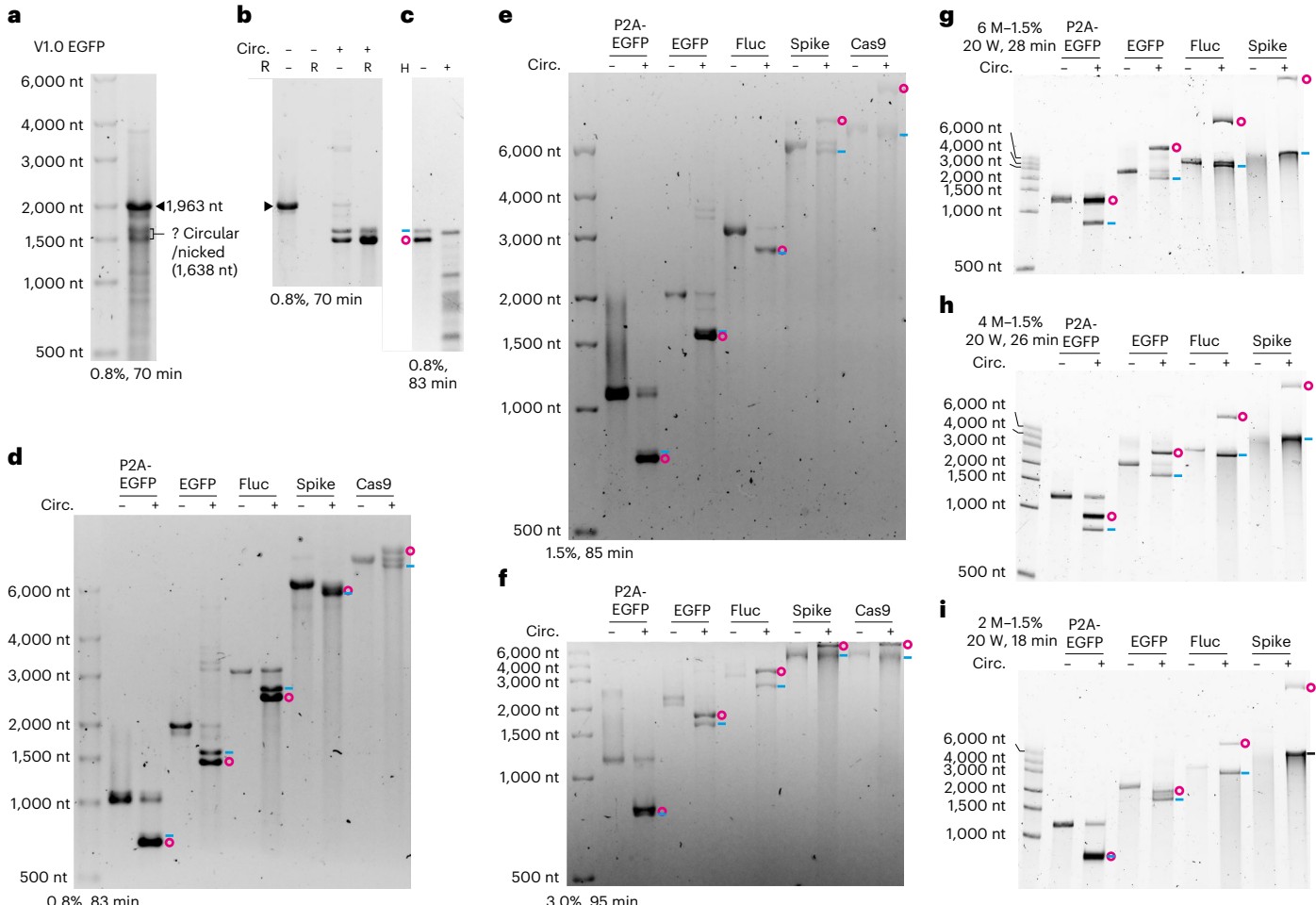

**Fig. 3 | Separation of circRNAs and their linear counterparts using agarose gels. a**, Cotranscriptionally circularized sample of the V1.0 EGFP analysed on a native agarose gel. The FL EGFP is 1,963 nt and the circEGFP is 1,638 nt. Two bands were observed around the circEGFP position. **b**, FL and circularized (circ.) EGFP were treated with RNase R (R) and analysed on a native agarose gel. **c**, RNA of the lower band in the gel from **b** was purified, RNase R digested and then linearized with RNase H (H) in the presence of a 34 nt DNA primer and analysed on a native agarose gel. **d–i**, FL and circularized RNAs analysed on 0.8% (**d**), 1.5% (**e**) and 3.0% (**f**) native agarose gels or 1.5% urea–agarose gels with 6 M (**g**), 4 M (**h**) or 2 M (**i**) urea. P2A-EGFP has no CVB3 IRES. Electrophoresis of urea–agarose gels was operated in a vertical system with specified settings.

band appeared above the Cas9-FL. This slow-moving band remained in the 1.5% gel and was also observed for the Spike, suggesting it is circRNA (Fig. 3e). The separation for the other circRNAs is reduced in the 1.5% gel, indicating that increasing agarose concentration more significantly decreases the mobility of circRNAs than that of linear RNAs. This was further confirmed using a 3% gel, where all circRNAs ran slower than their linear counterparts (Fig. 3f). Circularity of these slow-moving bands in the 3% gel was confirmed by RNase R digestion (Extended Data Fig. 3a). These results demonstrate that sufficient separation between circRNA and its linear counterpart is achievable in native agarose gels (see also Extended Data Fig. 4).

### The urea–agarose gel system provides excellent separation between circRNAs and their linear equivalents

Next, we tested denaturing agarose gels for separation of circRNAs and their linear forms. In formaldehyde agarose gels, we observed separation only for Spike and Cas9 in a 1.5% gel (Extended Data Fig. 3b,c). Due to its toxicity, formaldehyde can be substituted with urea[26]. Notably, in a 6 M–1.5% urea–agarose gel, all tested circRNAs moved drastically slower than their linear counterparts (Fig. 3g). Lowering the urea or agarose concentration reduced the separation (Fig. 3h,i and Extended Data Fig. 3d,e). In a 6 M–4% gel, even the 141 nt circ3×Flag ran slower than its nicked form (Extended Data Fig. 3f). We also observed that

extending the running time or increasing the running power enhanced the separation (Extended Data Fig. 3g–j). Notably, electrophoresis of these gels only takes <30 min, making it a rapid method for circRNA identification (Extended Data Fig. 4).

### The TRIC-V2 enables RNA circularization without unwanted sequences

An ideal TRIC construct should have minimal sequence requirements while remaining efficient (Fig. 4a). However, both V1.0 and PIE rely on native splicing sequences that will remain part of the final circR-NAs. To address this limitation, we reduced the tRNA sequence to retain only P1 and P10 (V1.30; Extended Data Fig. 5). However, this markedly reduced the circularization efficiency (Fig. 4b). Restoring the R30 (V1.33) recovered the efficiency to V1.0 level while restoring the L15 (V1.32) or introducing a 3' exon–EGS interaction (V1.31) did not. Further optimizations of the L/R lengths (V1.34–1.39) showed that the 17 nt ACA (V1.39) is the minimal sequence for achieving the V1.0 efficiency[25].

Since structures, rather than sequences, are crucial for group I intron activity[27], we next generated various V2 constructs that do not rely on native splicing sequences (Extended Data Fig. 5l–n) with altered anticodon stems (V2.0 and V2.1) or reversed P1 and P10, while keeping the U for the G•U base pair, essential for the splicing reaction (V2.2).

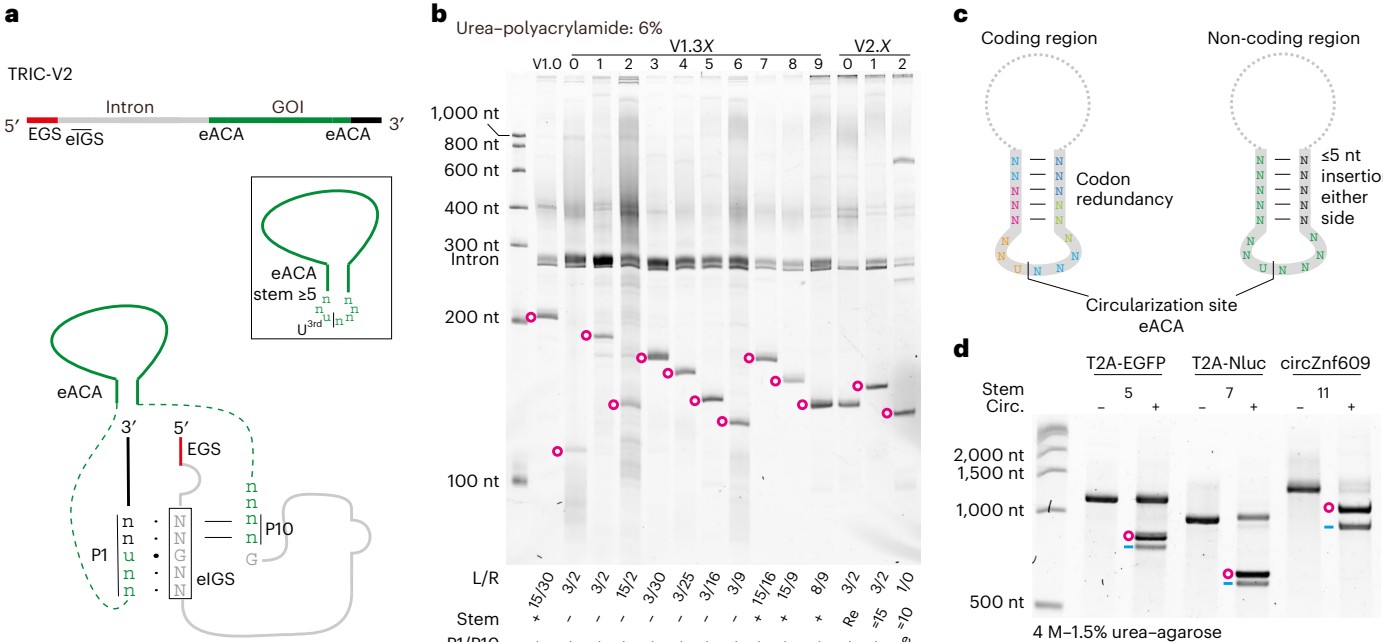

**Fig. 4 | TRIC-V2 enables circularization without unwanted sequences. a**, The concept of the TRIC-V2, which has minimal sequence requirements yet remains high efficiency. eIGS, extended IGS. **b**, The L15/R30 of the tRNA sequence in V1.0 were systematically engineered. For V1.30–39 (V1.3X), the tRNA sequences were gradually reduced. L8/R9 in V1.39 is the ACA of the tRNA. For V2.0–2.2 (V2.X), the sequences of the anticodon stem were reversed (re) (V2.0) or extended (15 bp, V2.1), or the sequences of P1/P10 were reversed (V2.2). Cotranscriptionally circularized samples were analysed on a 6% urea–polyacrylamide gel. **c**, An effective eACA should consist of a 7 nt loop with a U at the third position and a stem ≥5 bp. The circularization site can be placed either in a UTR or CDS. **d**, Circularization sites of the T2A-EGFP, T2A-Nluc and circZnf609 were placed in CDS with stem lengths of 5, 7 and 11 bp, respectively. V2 circZnf609 is identical to the natural circZnf609 except for nine non-sense mutations in CDS. FL of each RNA was circularized for 20 min and analysed on a 4 M–1.5% urea–agarose gel.

All constructs circularized efficiently (Fig. 4b), thus, V2 minimally requires an extended ACA (eACA) comprising a 7 nt loop with a U at the third position and a ≥5 bp stem for efficient circularization (Fig. 4c and Extended Data Fig. 6a). We then tested circularization of three longer protein coding circRNAs. Multiple eACAs were found in each coding sequence (CDS) and all GOIs were circularized efficiently (Fig. 4d). Due to the small size and sequence requirements of this eACA motif, suitable sites can be easily identified or, in case of protein CDS, engineered by making use of codon redundancy. Alternatively, short (<5 nt) insertions can be made in non-coding regions (Fig. 4c).

**The TRIC-V2 is faster than the PIE**

The circularization efficiency was highest for the natural circZnf609[28], which exhibits the longest stem (11 bp) among the three constructs (Fig. 4d), suggesting that longer stems lead to higher efficiency. To further investigate this, we generated V2 constructs with stem lengths of 15 and 25 bp. As expected, V2 (stem of 15 or 25 bp) outperformed V1.0 (stem of 5 bp) in circularizing EGFP (Fig. 5a). Notably, further extending the EGS to 40 nt did not obviously enhance the efficiency.

The superior efficiency of V2 over V1.0 suggests that V2 would probably outperform PIE as well. To compare V2 and PIE, we circularized EGFP, Spike and Cas9-FL for 1–4 min (Fig. 5b–d). V2 (stem of 25 bp) outperformed PIE in efficiency for all tested candidates. For instance, in 1 min, V2 (stem of 25 bp) converted over half of the FL into circEGFP, while PIE only converted a small portion (Fig. 5b). The higher efficiency is reflected by the construct kinetics: using the Michaelis–Menten analysis[22], we found that $V_{max}$ of the V2 (stem of 25 bp) is 3.7 times higher than the PIE $V_{max}$ in the EGFP circularization (Fig. 5e and Extended Data Fig. 6b–q). V2 (stem of 25 bp) showed higher efficiency compared with V2 (stem of 15 bp), suggesting potential space for further efficiency improvement. Additionally, PIE generated a substantial number of concatenations, which were mostly absent from V2 (Fig. 5b).

Next, we compared EGFP circularization efficiency, yield and nicking between EGFP V2 (stem of 25 bp) and PIE constructs (Fig. 5f and Extended Data Fig. 5r). In 3 min, V2 (stem of 25 bp) achieved slightly higher efficiency and yield than PIE in 8 min, with significantly less nicking. Extending the circularization time to 8 min improved V2 (stem of 25 bp) to 97.8% efficiency and 74.8% yield (90.3% of the yield limit), but also increased nicking. Notably, using the FL post-transcriptional circularization protocol, PIE improved yield and nicking from previously reported values of ~50% and ~20% to 65.5% and 9.3%, respectively[6]. Further yield enhancement from 65.5% (PIE) to 74.8% (V2) should be attributed to the higher circularization rate of V2.

Recent studies have described *Tetrahymena thermophila* (Tetra) group I intron-derived constructs, the Tetra-STS[29] and the Tetra-Rzy[30], for circRNA synthesis. We compared these with V2 by cloning CVB3-EGFP into Tetra-STS (AU-rich no. 16) and Tetra-Rzy (CVB3 IRES-GFP) constructs. As shown in Fig. 5g,h, V2 outperforms both constructs. Additionally, the Tetra-V2 also produced circCVB3-EGFP efficiently, demonstrating that optimizations from the Ana intron can be effectively applied to other group I introns.

**Immunogenicity of circRNAs is low**

The immunogenicity of circRNAs remains debatable[8,11,16,31,32]. Recent research suggested that PIE-derived residual bacterial sequences cause circRNA immunogenicity[16]. Given that V2-derived circRNAs lack bacterial sequences, they might be less immunogenic than PIE circRNAs. To test this, we designed V2 CVB3-EGFP and CVB3-Nano luciferase (Nluc) constructs, where the eACA is either formed by the CDS (EGFP, stem of 5 bp) or derived from the 25ES7b stem of human 28S ribosomal RNA (rRNA) (Nluc, stem of 24 bp). CircRNAs were purified by three consecutive high-performance liquid chromatography (HPLC) runs followed by RNase R digestion[8] (HR purification; Fig. 6a). Subsequently, we transfected these circRNAs and controls into A549 (human lung carcinoma)

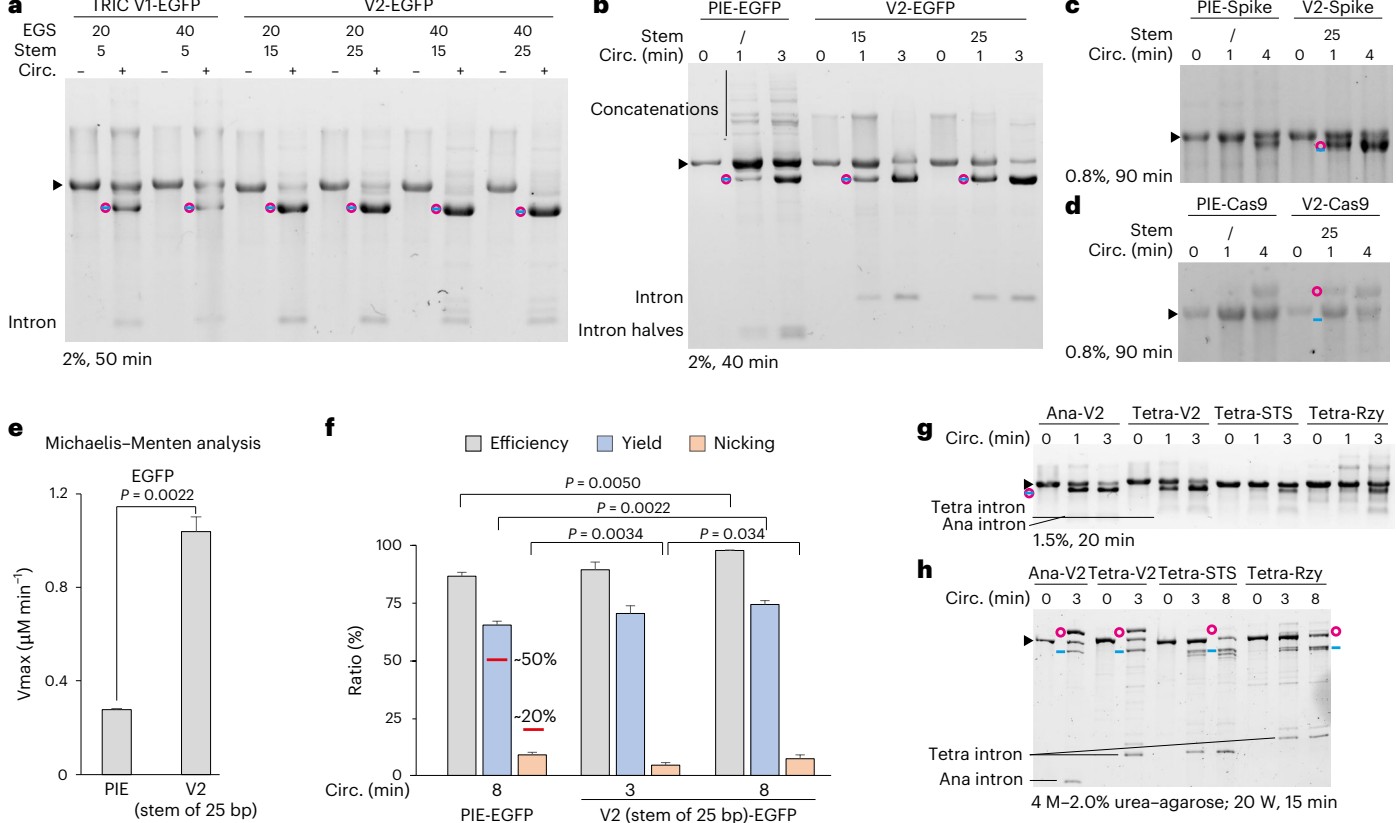

**Fig. 5 | TRIC-V2 is faster than the PIE. a**, Comparison between V1.0 and V2 in circularizing EGFP. EGSs were either 20 or 40 nt. The eACA stem for V2 constructs was either 15 or 25 bp. FL of V1.0 and V2 were circularized for 4 min and analysed on a 2% native agarose gel. There was no separation between circRNA and its linear form in 2% gels. **b**–**d**, The comparison between V2 (stem of 25 bp) and PIE for circularizing EGFP (**b**), Spike (**c**) and Cas9 (**d**). FL were circularized for 1–4 min. **e**, Michaelis–Menten analysis of the V2 (stem of 25 bp) and the PIE for circularizing EGFP. Calculations based on the reaction processes observed in native agarose gels (Extended Data Fig. 6b–q). **f**, FL of V2 (stem of 25 bp) EGFP and PIE EGFP were circularized for either 3 and 8 min or 8 min and analysed on a native gel (Extended Data Fig. 6r). The intensities of the remaining FL, circRNA and nicked RNA were measured in ImageJ. Efficiency was calculated by dividing the intensity of spliced FL by the total FL, yield was determined by the intensity of circRNAs divided by total FL and nicking was calculated as the ratio between nicked RNAs and the sum of circRNA and nicked RNA. The red lines indicate the reported values of yield and nicking for a circEGFP[6]. **g**,**h**, The comparison between V2 using either the Ana group I intron or the Tetra group I intron and the Tetra-STS and the Tetra-Rzy constructs: CVB3 (coxsackievirus B3)-EGFP was cloned to the best Tetra-STS (AU-rich no. 16) and Tetra-Rzy (CVB3 IRES-GFP) constructs, and FL were circularized for the designated time and analysed on native agarose gels (**g**) or urea–agarose gels (**h**). The data in **e** and **f** are mean ± s.d. for three independent replicates tested using unpaired two-tailed *t*-tests.

cells and monitored expression of RIG-I (5′-triphosphate sensor) and several cytokines[33]. As expected, poly(I:C) (double-stranded RNA mimic) and unmodified mRNAs substantially increased RIG-I and cytokine expression compared with the lipofectamine-only control (Fig. 6b). CircRNAs exhibited lower immunogenicity compared with unmodified mRNAs, but still induced notable RIG-I and cytokine expression. We consistently observed reduced immune responses with V2 circNluc compared with PIE circNluc, but not for V2 circEGFP. Therefore, the immunogenicity comparison between V2- and PIE-derived circRNAs remains inconclusive.

RNase R might enrich structured linear RNAs that are immunogenic[17]. We therefore moved the RNase R digestion before HPLC (RH purification; Extended Data Fig. 7a). Subsequent tests indicated comparable immunogenicity of circRNAs generated by the HR and RH purifications (Extended Data Fig. 7b). However, the V2 circNluc elicited heightened RIG-I expression, suggesting 5′-triphosphate contamination. Thus, we further added an alkaline phosphatase treatment before RNase R (PRH purification; Fig. 7c and Extended Data Fig. 7c). The phosphatase-treated unmodified RNAs indeed exhibited a significantly reduced immunogenicity[11,16]. Remarkably, all PRH-purified circRNAs failed to induce RIG-I or cytokine expression compared with the lipofectamine only (Fig. 6d). An exception was noted for V2 circEGFP, which induced minimal yet significant CCL5 expression (Fig. 6d, insert). This

might be due to FL contamination resulting from the low circularization efficiency (stem of 5 bp). In conclusion, these results support the notion that circRNAs have low immunogenicity[8,32].

### Synthesis and immunogenicity of modified circRNAs

The significant residual immune response to phosphatase-treated linear mRNAs highlights the potential immunogenicity of nicked circRNAs (Fig. 6d), which could be diminished by RNA modifications[34–36]. However, the PIE, and probably the TRIC as well, are unable to synthesize fully modified circRNAs since modifications could disrupt the ribozyme activity[8]. Since group I introns can also function as *trans*-excision ribozymes (TER)[37–39] (Figs. 1e and 6e), relocating the 3′ and 5′ parts of the bridge to 5′ and 3′ ends of a GOI, respectively, would allow circularization of the GOI in *trans* (Figs. 1f and 6f and Extended Data Fig. 8a). With this *trans*-excision ribozyme-based circularization (TERIC), modified GOIs can be circularized by unmodified TERs.

To test this idea, the IGS and 5′ half of the P9.0 were retained in TERs to create P1/P10 and P9.0 interactions between TERs and GOIs[38,39]. Additionally, the key elements for the TRIC were also preserved (Extended Data Figs. 1 and 8a). Initial tests showed that the TER(1,226) circularized unmodified EGFP efficiently (Fig. 6g and Extended Data Fig. 8b,c). The circularization was confirmed and further optimized (Extended Data Fig. 8d–h). Neither PIE nor TRIC or TERIC produced

100% $N^6$-methyladenosine (m⁶A) or $N^1$-methylpseudouridine ($N^1\Psi$) modified circRNAs (Extended Data Fig. 9). We realized that these modifications could weaken the AU-rich IGS of TER(1,226) and thus additionally analysed circZnf609, whose IGS is CCGCC. As expected, TERIC synthesized 100% m⁶A and $N^1\Psi$ modified circZnf609 efficiently, but not PIE or TRIC (Fig. 6h). Of note, precursors containing $N^1\Psi$ were circularized more efficiently than m⁶A-modified ones. In summary, TERIC enables efficient synthesis of modified circRNAs.

Next, we tested immunogenicity of the circZnf609 with or without modifications (Fig. 6i,j). Consistently, HR-purified, but not PRH-purified, unmodified circRNAs exhibited significant immunogenicity, regardless of the source (V2 or PIE). However, the modified circZnf609 did not show significant advantages over PRH-purified unmodified circRNAs[8]. It is possible that longer transfection times or higher doses are required to observe potential differences. In contrast, m⁶A-modified circZnf609 induced significant RIG-I expression.

### TRIC circNluc produces more proteins than PIE circNluc

To study the translation efficiency of TRIC circRNAs, we investigated protein expression from circRNAs containing a CVB3 IRES. In A549 cells, Nluc expression from $N^1\Psi$-modified mRNA peaked at day 1 and then declined rapidly, whereas expression from V2 circNluc peaked at day 2 and remained significantly higher than day 1 up to day 3 (Fig. 7a). Similar patterns were seen in HEK 293F cells (Extended Data Fig. 10a,b). Meanwhile, PIE circNluc expressed similarly to V2 circNluc on day 1, but showed no increase afterwards. Consequently, V2 circNluc led to higher expression after day 2 (Fig. 7a,b). On day 7, V2 circNluc expression remained at almost half the level of that from $N^1\Psi$-modified mRNA on day 1.

### Translation efficiency of RCT is increased by over 7,000-fold

Unlike single-shot translation, RCT primarily produces a polyprotein. This can be converted into protein monomers provided with 2A skipping sequences[40] (Fig. 7c,d). Instead of using circRNAs with only a CDS, which are inefficient in translation initiation[9,10,40], we selected a potent short IRES (OR4F17)[40] for the RCT construct. As expected, the circOR4F17-Nluc-RCT produced a significant number of proteins. However, the translation was still orders of magnitude less efficient compared with $N^1\Psi$-modified mRNA (Fig. 7f).

We then asked whether potent viral IRESs could be used for RCT. However, viral IRESs are typically long and highly structured, which could block RCT either by in-frame stop codons or by causing ribosome stalling. Nevertheless, we tested the 373 nt CSFV IRES[6,40,41] (Fig. 7e). As expected, multiple stop codons were identified in each frame. We therefore engineered frame 1 to eliminate stop codons and align with the CDS. Notably, the RCT using CSFV IRES showed over a tenfold increase in Nluc expression compared with its single-shot translation and was over 7,000-fold more efficient than the RCT using the OR4F17 IRES (Fig. 7f and Extended Data Fig. 10c).

## Discussion

Synthesis of long circRNAs faces challenges due to low yields[6]. Here, we show that by suppressing the cotranscriptional circularization and increasing the post-transcriptional circularization rate using the TRIC constructs, high yields for long circRNAs are achievable. For instance, V1.0 (a version with comparable circularization kinetics to PIE), can efficiently generate the 8,706 nt Factor 8 circRNA and V2 achieves a 90.3% yield for the 1,638 nt circular EGFP. Furthermore, V2 enables circularization without unwanted sequences, which is crucial for natural circRNAs or circRNAs for RCT. Recent studies have reported Tetra group I intron-derived constructs for clean RNA circularization[29,30]. However, these constructs resemble the V1.30 (Fig. 4b and Extended Data Fig. 5), exhibiting low in vitro efficiency.

One unexpected observation is that native agarose gels effectively separate circRNAs from their linear counterparts beyond 8,000 nt. Previous studies probably missed this separation because commonly studied circRNAs (~1,500 nt) run similarly to their linear forms in the typically used 1–2% gels with short running times. Urea was not considered an effective denaturing reagent in the agarose gel system[42]. However, we demonstrate that urea–agarose gels offer a simple, robust and tuneable method for separating circRNAs from their linear forms in the range of 140–6,000 nt. Taking the urea–polyacrylamide gels for short circRNAs into account, circRNAs of varying lengths can be effectively distinguished from their linear counterparts (Extended Data Fig. 4)[4].

Initially, we anticipated V2 circRNAs to be less immunogenic than PIE circRNAs due to their lacking residual bacterial sequences[16]. However, no detectable immune responses are observed from all PRH-purified circRNAs (including PIE-generated ones), confirming the low immunogenicity of circRNAs[8,32]. HR- or RH-purified V2 circNluc displays lower immunogenicity compared with the PIE circNluc. This reduction may result from less full-length contamination due to V2 efficiency or the use of a section of human 28S rRNA in the construct, which was used on the grounds that the innate immune system would have evolved not to react to that sequence. Consistent with the rRNA sequence and the absence of bacterial sequences playing a role, the V2 circNluc outperforms the PIE circNluc in the 7 day expression experiment, despite their similar 1 day immunogenicity. Modified circRNAs have comparable immunogenicity to unmodified circRNAs in the 1 day test[8]. However, in long-term applications, modifications may reduce immunogenicity arising from nicked species. Additionally, modifications have the potential to enhance stability and facilitate endosomal escape[43], making TERIC a valuable tool for circRNA applications.

Finally, we could show that the trans-splicing-based circularization methods and use of viral IRESs have now removed barriers for generating efficient RCT constructs. Notably, the CSFV IRES enhances the RCT efficiency by over 7,000 times, demonstrating the ribosome's ability to read through complex viral IRESs. RCT might provide additional

**Fig. 6 | CircRNAs have low immunogenicity. a**, CircRNAs were purified using HR, with circularized RNA purified by three consecutive HPLC runs and a final RNase R digestion. Late circRNA fractions from each injection were collected and 50 ng of each purified circRNA was analysed on a 6 M–1.5% urea–agarose gel. **b**, For circRNAs purified using HR, 100 ng of each purified circRNAs, poly(I:C) (double-stranded RNA mimic) and both modified and unmodified linear (Lin.) mRNAs (capped and poly-A tailed) were transfected into 10,000 A549 (human lung carcinoma) cells. The mRNA levels of RIG-I and several cytokines were monitored by RT–qPCR 24 h later. The y axes represent fold changes compared with the mock. **c**, CircRNAs were purified using PRH purification, with circularized RNA treated with alkaline phosphatase and RNase R before HPLC injection. Late circRNA fractions from each injection were collected and 50 ng of each purified circRNA was analysed on a 6 M–1.5% urea–agarose gel. **d**, CircRNAs purified using PRH were transfected and and monitored as in **b**. Insert: CCL5 expression of mock, lipofectamine only (lipo.) and circRNAs.

**e**, Group I introns function as TER to remove a middle (bridge) sequence from their target sequences. **f**, The TERIC produces circRNAs. **g**, Circularization of EGFP using TERIC were conducted in 10 µl volumes with 0.1, 0.5 or 2 mM GTP. In protocol A, in vitro transcribed TERs and pGs were initially mixed at a 5:1 ratio, refolded, supplemented with splicing buffer and heated at 55 °C for 20 min. Subsequently, splicing reactions were halted using 2 µl of 100 mM EDTA and analysed on a 0.8% native agarose gel. **h**, FL and 20 min circularized circZnf609 analysed on a native agarose gel. FL were synthesized either with or without 100% m⁶A or $N^1\Psi$ modification (Modi). TER was unmodified. **i**, CircRNAs were purified using either HR or PRH purification, as in **a** and **c**. **j**, CircRNAs purified using either HR or PRH purification were transfected and and monitored as in **b**. IFN-α, interferon-α; IFN-β, interferon-β; TNF-α, tumour-necrosis factor-α; IL-6, interleukin-6; CCL5, chemokine (C-C motif) ligand 5; AP, alkaline phosphatase; R, RNase R. Data in **b**, **d** and **j** are mean ± s.d. for three biological replicates. *$P < 0.05$, **$P < 0.01$ and ***$P < 0.001$, unpaired two-tailed $t$-test.

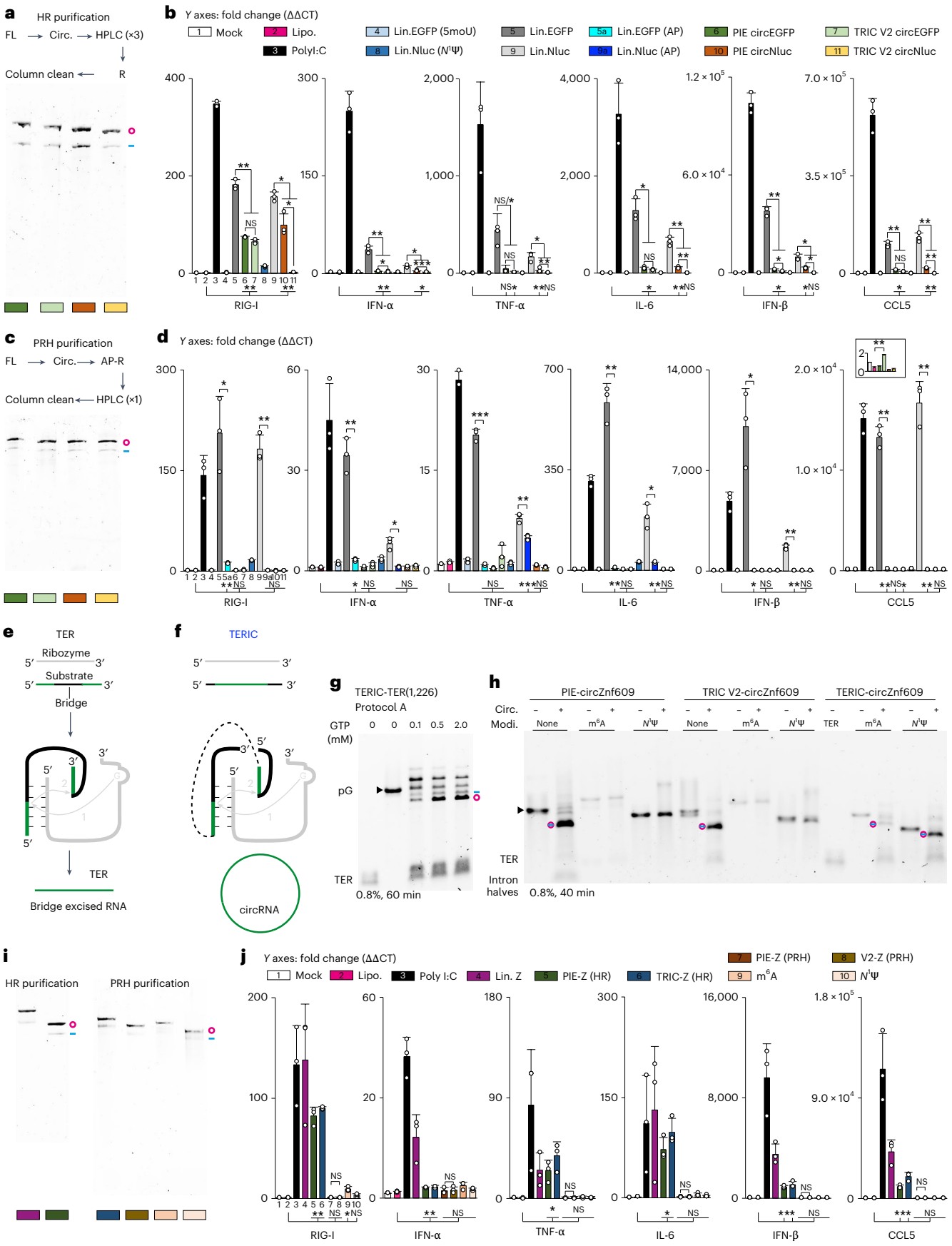

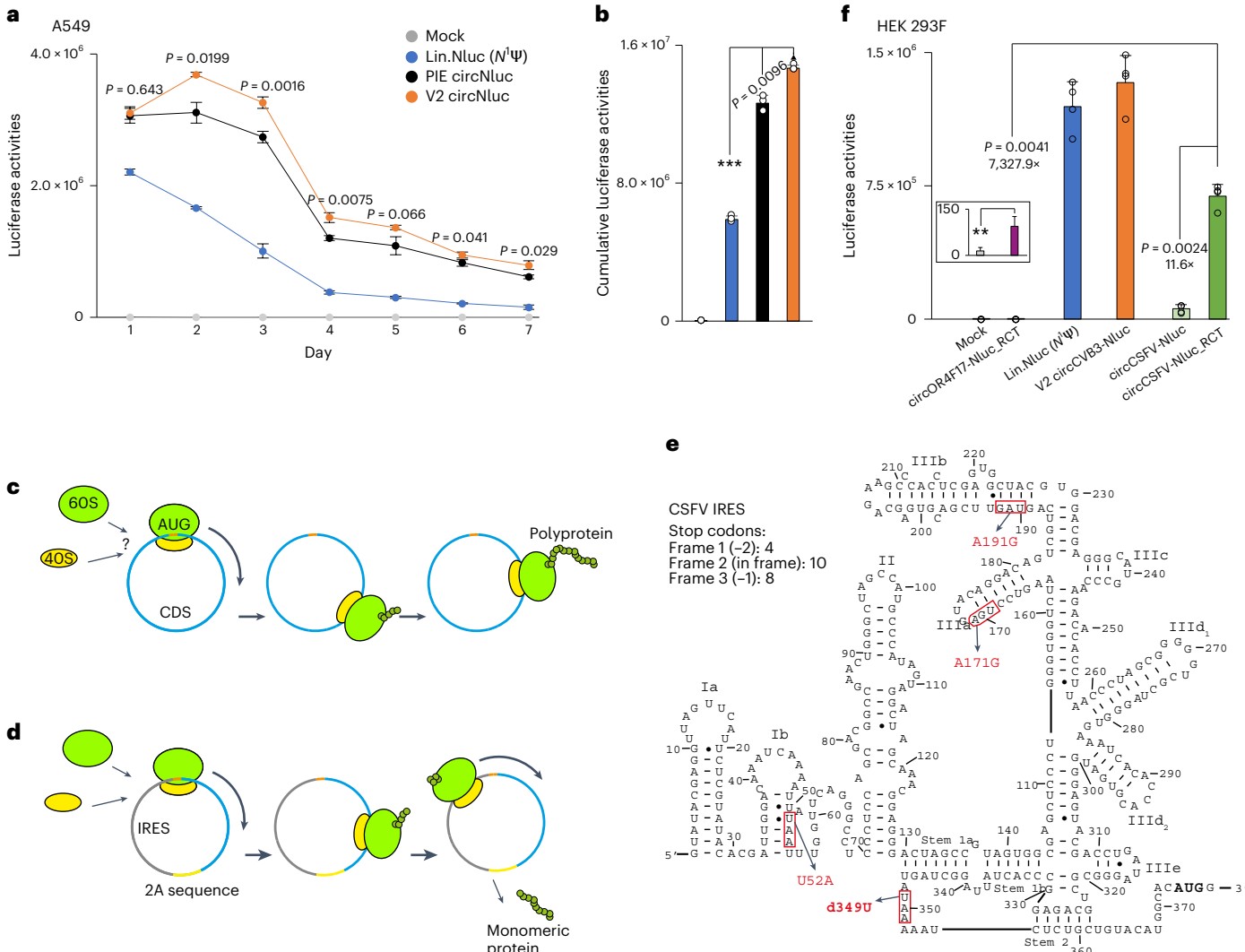

**Fig. 7 | Protein expression from circRNAs. a**, Equimolar amounts of $N^1\Psi$-modified linear Nluc, V2 circNluc and PIE circNluc were transfected into 10,000 A549 cells. Nluc expression was measured over 7 days. **b**, The cumulative expression of Nluc from $N^1\Psi$-modified linear Nluc, V2 circNluc and PIE circNluc. **c**, When circRNAs contain only a CDS but no stop codons, ribosomes can translate such circRNA indefinitely and produce a polyprotein, a process known as the RCT. **d**, When a 2A sequence is present, such as T2A or P2A, the polyprotein will be converted to protein monomers. **e**, The secondary structure of the 373 nt

CSFV IRES. Multiple stop codons are present in all possible frames: 4, 10 and 8 in frames 1–3, respectively. Frame 1 was cleaned of stop codons by three mutations (U52A, A171G and A191G), and one deletion (d349U) to align with the CDS. **f**, Equimolar amounts of circOR4F17-Nluc-RCT, circCSFV-Nluc, circCSFV-Nluc-RCT, V2 circNluc and $N^1\Psi$-modified linear Nluc were transfected into HEK 293F cells and 24 h later, Nluc expression was monitored. The data in **a**, **b** and **f** are mean ± s.d. for three (**a** and **b**) or four (**f**) biological replicates. ***$P < 0.001$, unpaired two-tailed $t$-test.

circRNA stabilizations by active translations. However, it will be important to characterize the IRES-derived translation products in future studies (Extended Data Fig. 10c). Although RCT efficiency using CSFV IRES is 52.1% of that from $N^1\Psi$-modified mRNA, integrating other efficient IRESs such as CVB3 and HRV-B3 could enhance RCT efficiency to many times higher than $N^1\Psi$-modified mRNAs[7].

## Methods

### Cloning

All sequences for IVT were cloned into a pUCIDT vector between a T7 promoter and an enzyme cleavage site (EcoR V or Not I+EcoR V). The TERIC sequences were retained as PCR-amplified fragments. The PIE and Tetra constructs were constructed as previously described[6,29,30]. Fluc, Spike and Cas9 genes were kindly provided by Nathan James, Kat Ciazynska and Jerome Zurcher, respectively. The Factor 8 gene was obtained from Addgene (41036). The remaining sequences were synthesized by Genewiz or Integrated DNA Technologies,

respectively. The Gibson assembly (New England Biolabs (NEB)) was used to assemble DNA fragments with PCR linearized vectors and the QuickChange technique using Q5 DNA polymerase (NEB) was employed for all insertions, deletions, or mutations. Plasmids were amplified using homemade TOP10 competent cells and verified by Sanger Sequencing (Genewiz). Supplementary Table 1 lists all primers and plasmids.

### DNA templates for IVT

Plasmids for IVT were amplified in TOP10 cells grown in a 150 ml Lysogeny broth (LB) culture and subsequently purified using the Maxi Plus plasmid purification kit (Qiagen). The resulting plasmids were linearized using either EcoR V HF (NEB) or Not I (NEB) and then deproteinized using phenol:chloroform:isoamyl alcohol (Sigma) extraction (PCI) and ethanol precipitation. 2′-O-methylated 5′ reverse primer was used to amplify the template for the Tetra-STS construct as previously described[29,30]. The TERIC templates were generated by PCR followed by PCI purification.

## RNA synthesis

The 5-methoxyuridine-modified linear EGFP mRNA was purchased from TriLink (L-7201). Homemade linear EGFP or linear Nluc bear the same untranslated regions (UTRs) as in the Pfrizer/BioNTech COVID-19 vaccine[44]. RNAs were synthesized in 100–500 μl IVT reactions using the following recipe: 50 μg ml$^{-1}$ of DNA template, 14 μg ml$^{-1}$ of homemade T7 polymerase, 40 U ml$^{-1}$ of RNase inhibitor (Promega), 6 mM of each NTP, 1 U ml$^{-1}$ of pyrophosphatase (Thermo Fisher Scientific) and 1× IVT buffer. For IVT involving modified RNAs, ATP or UTP was completely replaced with m$^6$ATP (Jena Bioscience) or $N^1$ΨTP (Jena Bioscience), respectively. For IVT of linear mRNAs or situations allowing cotranscriptional circularization, 1× IVT buffer contained 80 mM Tris–HCl (pH 7.4), 2 mM spermidine, 40 mM dithiothreitol and 24 mM MgCl$_2$. To assess the impact of Mg$^{2+}$ on cotranscriptional circularization and IVT yield, Mg$^{2+}$ concentrations in 1× IVT buffer ranged from 6 to 24 or 10 to 24 mM. IVT yield was given by microgram of RNA produced per microliter of IVT. For suppressing cotranscriptional circularization, the concentration of Mg$^{2+}$ in 1× IVT buffer was set to 16 mM. IVT reactions were incubated at 37 °C for 3–5 h and subsequently treated with RNase-free DNase I (NEB) for 20 min. Following this, an equal volume of 7.5 M lithium chloride (Sigma) was added to precipitate RNAs for 30 min to overnight at −20 °C. The precipitates were then collected by centrifugation at 13,000 rpm per min for 20 min. RNA pellets were washed with 75% ethanol, air dried and dissolved in diethyl pyrocarbonate (DEPC)-treated H$_2$O. Linear mRNAs were later capped using the Vaccinia capping system (NEB) and polyadenylated by the *E. coli* poly-A polymerase (NEB).

## RNA circularization

For situations where cotranscriptional circularization is allowed, the post-transcriptional circularization involved supplementing DNA template digested IVT reactions with an additional 2 mM of GTP and heating at 55 °C for 20 min[6]. In cases where cotranscriptional circularization was suppressed, 9 μl of FL was firstly refolded by heating at 95 °C for 2 min and then cooling on ice for 3 min. The refolded RNAs were supplemented with 1 μl of 10× circularization buffer (500 mM Tris–HCl, pH 7.4, 100 mM MgCl$_2$, 10 mM dithiothreitol and 20 mM GTP) and heated at 55 °C for specified times. The final FL concentration was 0.5 μM for Spike, Cas9 and Factor 8 and 1 μM for the rest. After incubation, 2 μl 100 mM EDTA was used to stop the circularization. All samples were analysed by gels, and the intensity of the RNAs of interest was quantified using ImageJ. The reactions for analysis of circularization efficiency, yield, nicking and kinetics were independently conducted three times. Particularly, circRNAs for immunogenicity studies or protein expression were circularized in 750 μl volumes for an initial time of 20 min. When the remaining FL was significant, the reaction was then subjected to an additional 20 min of circularization.

In the case of TERIC, two protocols were employed. In protocol A, TERs and precursor of GOIs (pGs) were mixed and refolded in DEPC-treated H$_2$O. Subsequently, they were supplemented with splicing buffer and heated at 55 °C for 20 min. In protocol B, the refolding of TERs and pGs occurred separately in the splicing buffer, followed by mixing and 20 min of circularization at 55 °C. Splicing conditions, including the concentration of GTP, the TER to pG ratio, the concentration of pGs and the reaction duration, were optimized.

## RNA purification

For gel-based purification, circularized RNAs were initially separated using either urea–polyacrylamide gels or native agarose gels. Subsequently, the target RNAs were either electroeluted (urea–polyacrylamide gels) and column purified or extracted (native agarose gels) using a gel extraction kit (Zymo Research) from the excised gel bands.

In the case of HPLC-based purifications, 5–50 μg of each RNA in 100 μl DEPC H$_2$O was first refolded as described above and then injected into a gel filtration column (Sepax, 215980P-4630), operated on an AKTA Pure Micro system (Cytiva) without UV illumination. The running buffer contained 10 mM Tris–HCl (pH 6.5) and 1 mM EDTA. Fractions were analysed using native agarose gels. Target fractions were then pooled and spin column cleaned (Zymo Research). For linear mRNAs, spin column purification after HPLC was the final step. For the HR purification, three consecutive HPLC injections were performed, during which the late circRNA fraction of each injection was collected as previously described[8]. The late circRNA fraction from the final HPLC run was subjected to a 1 h RNase R digestion (antibodies-online), followed by spin column purification (Zymo Research). For RH purification, circularized RNAs were digested with RNase R for 1 h before injection. For PRH purification, circularized RNAs were first treated with alkaline phosphatase (Invitrogen) for 1 h, followed by a 1 h RNase R digestion, and then injected into the HPLC column. Circularized V2 CVB3-EGFP (stem of 5 bp) was injected to HPLC for a pre-purification before the RH or PRH purification to reduce FL contaminations.

## Cell culture and transfection

HEK 293F cells (Thermo Fisher Scientific) were cultured in Freestyle media (Gibco) in a shaker operated at 37 °C, 125 rpm per min and 8.5% CO$_2$ (INFORS HT). A549 cells (American Type Culture Collection, CCL-185) were cultured in Dulbecco's modified Eagle medium with 10% FBS (High Glucose GlutaMAX, Life Technologies Ltd) in a Sanyo incubator at 37 °C with 5% CO$_2$.

For EGFP or Nluc expression, 0.04 pM of each RNA was transfected into 10,000 cells in 100 μl in 96-well plates (Corning) using the MessengerMax transfection reagent (Thermo Fisher Scientific). Protein expression levels were measured from day 1 to 7. For A549 cells, media were renewed every day, and for the HEK 293F cells, 15 μl of media was topped up each day to compensate for evaporation.

For immunogenicity studies, 100 ng of poly(I:C) (Merck) or every other RNA was transfected into 100,000 A549 cells in 500 μl media in 24-well plates (Corning) using MessengerMax transfection reagents. At 24 h post-transfection, cells were lysed and total RNAs were extracted using TRIzol (Invitrogen) and the RNA Clean & Concentrator kit (Zymo Research). Total RNAs were eluted in 10 μl DEPC-treated water and stored at −80 °C for quantitative RT–PCR (RT–qPCR).

## RT–PCR

For RT–PCR, initially, 10 ng of each RNA was reverse transcribed in a 20 μl reaction volume using SuperScript IV Reverse Transcriptase (Thermo Fisher Scientific) with reverse transcription (RT) primers listed in Supplementary Table 1. Next, 2 μl of each RT product was used as a template for a 50 μl PCR reaction (Q5 DNA polymerase, NEB). To verify the circularization junction, PCR products were cloned to the pUCIDT vector and Sanger sequenced. Forward primers of EGFP, Fluc, Spike, Cas9 and Factor 8 were designed to prime in the CDS of each gene (Supplementary Table 1).

For RT–qPCR, 300 ng of total RNA for each sample was reverse transcribed using SuperScript IV with a primer mixture containing random hexamers and oligo dT (Thermo Fisher Scientific). Subsequently, 4 μl of each 20-fold diluted RT product was mixed with 6 μl of enzyme mixes in a 384-well plate (Applied Biosystems). Three replicates were used for each target in each sample. The enzyme mixes were prepared using the iTaq Universal SYBR Green Supermix (Bio-Rad) and the primers listed in Supplementary Table 1. qPCR was conducted using a ViiA 7 Real-Time PCR System (Thermo Fisher Scientific) following a standard comparative CT protocol. Data analysis was carried out using the Quantstudio Real-time PCR software v1.3.

## RNase R/H digestion

RNase R digestion was conducted with 0.5 U of RNase R (antibodies-online) per 1 μg RNA concentration at 37 °C for the specified duration. For RNase H digestion (NEB), 5 μg of gel-extracted circRNA was first annealed with a fivefold molar amount of primer by heating at 65 °C

for 5 min, followed by 3 min of cooling on ice. RNase H and its buffer were then added for digestion at 37 °C for 20 min.

### Gel analysis

Urea–polyacrylamide gels and formaldehyde agarose gels were prepared and operated as previously described[14,45]. For 0.8–3.0% native agarose gels, the gels were prepared with DEPC-treated $H_2O$, using 1× TBE (89 mM Tris, 89 mM boric acid and 3 mM EDTA) as buffer, and with the addition of SYBR Safe or SYBR Green gel stains (Thermo Fisher). Gels were run horizontally in an Owl Easy Cast B2 Mini Gel Electrophoresis Systems (Thermo Scientific) at a constant power of 25 W for designated times.

For the preparation of urea–agarose gels, agarose was initially boiled in DEPC-treated $H_2O$, mixed with urea and then topped up with DEPC $H_2O$ and 10× TBE to achieve a final 1× TBE concentration. Agarose and urea amounts were scaled accordingly. The gels were prepared with 1.5 mm spacers and left at 4 °C for gelation for at least 4 h. Electrophoreses were operated in a vertical system (Bio-Rad, Mini-Protean Tetra Vertical Electrophoresis Cell system). To prevent gels from slipping during running, a 0.75 mm bottom spacer was attached to the gel plate. For the 6 M–4% urea–agarose gel, gel plates were preheated to prevent gelation during gel pouring. Gels were run at room temperature at 15–25 W for 15–30 min.

For loading of circularized samples, 50 ng (urea–agarose gels), 100 ng (urea–polyacrylamide gels) or 500 ng (other agarose gels) of each RNA sample was first denatured in urea loading buffer (NEB) or formamide loading buffer (Thermo Fisher) at 95 °C for 2 min and then loaded onto each gel. In some cases, a lower amount of FL RNA was loaded onto each gel to minimize FL dimerization. Urea–polyacrylamide gels and urea–agarose gels were stained with SYBR Green (Thermo Fisher Scientific) in 1× TBE for 10 min. All gels were imaged using a Bio-Rad ChemiDoc XRS+ Imaging System.

### Analysis of circularization efficiency, yield, nicking and construct kinetics

Circularization efficiency was calculated by dividing the intensity of estimated spliced FL by the total FL. The estimated spliced FL equals the sum of circular and nicked RNAs divided by the molecular weight ratio between FL and circRNA. The total FL is the sum of estimated spliced FL and remaining FL. Products of first step splicing, when visible, were assumed to be remaining FL. The yield was determined by the intensity of circRNAs divided by total FL, and nicking was calculated as the ratio between nicked RNAs and the sum of circRNA and nicked RNA. Yield limit is the yield when FL is completely converted to circRNAs with no nicking. For the efficiency determination of V1.0, the L15/R51 of tRNA^Leu, the internal homology arm and the spacers as used in the PIE were introduced into V1.0 so that V1.0 and PIE generate identical circRNAs for each GOI. Circularization was carried out for 10 min for Factor 8 and 20 min for the remaining GOIs. Initially, all samples were loaded onto a 0.8% native agarose gel and electrophoresed for 80 min of running at 25 W constant power. There was no separation between circular and nicked RNAs for the Spike and FL and nicked RNAs for the Factor 8. Therefore, the same gel was subjected to an additional 40 min of running to separate FL and nicked RNAs for Factor 8. The Spike samples were loaded onto a 1% native agarose gel for an 80 min run. To analyse the circularization efficiency of V2 and PIE, circularization was carried out for 3 and 8 min for the V2 (stem of 25 bp) and 8 min for the PIE. All samples were loaded onto a 0.8% native agarose gel and run for 80 min at 25 W.

Construct kinetics based on the Michaelis–Menten equation were performed as previously described[22]. Circularization reactions were carried out in 1× splicing buffer containing various GTP concentrations (1, 2.5, 5, 25, 100, 500, 1,000 and 2,000 µM) at 55 °C for four to six specified timepoints ranging from 0.5 to 15 min. The reactions were loaded onto 2% native agarose gels for 40 min of running at 25 W. The remaining ratio of FL was estimated by dividing the intensity of remaining FL

by the total FL. A semi-log plot correlating remaining ratio and time was utilized to estimate $t_{1/2}$ and $V_{obs}$ ($V_{obs} = 0.693/t_{1/2}$). Subsequently, the $V_{obs}$ and GTP concentration were plotted according to the Michaelis–Menten model, enabling the calculation of $V_{max}$ for the TRIC-V2 (stem of 25 bp) and the PIE. All source data are listed in Supplementary Table 2.

### Protein expression

GFP expression in HEK 293F cells was visualized using the EVOS FL microscope (Life technologies) at the designated time. The Nluc assay was carried out following the manufacturer's protocol (Promega). In brief, HEK 293F and A549 cells were initially lysed (25 mM Tris–HCl, pH 7.4, 150 mM NaCl, 10 mM $MgCl_2$, 1 mM EDTA, 2% glycerol and 1% Triton X-100) either by mixing 5 µl of culture with 495 µl of lysis buffer (HEK 293F) or by adding 100 µl of lysis buffer to each media-removed well (A549). After 10 min of incubation at 37 °C, 5 µl of each cell lysate was mixed with 5 µl of freshly prepared assay reagent in a black 384-well plate (Greiner). Three minutes later, luciferase activity was measured using a Spark 10 M plate reader (Tecan) with a 1,000 ms integration time.

### Western blotting

We found that antibodies against Nluc were not sensitive enough to detect expression from corresponding circRNAs in cells. Therefore, a 3×Flag tag was fused to the N-terminal of Nluc in the CSFV-Nluc and CSFV-Nluc-RCT constructs. One day after cell transfection, 10 µl of CSFV-flag-Nluc and CSFV-flag-Nluc-RCT circRNA transfected HEK 293F cells were collected and suspended in SDS lysis buffer (10 mM Tris, pH 8.0 and 1% SDS) and then mixed with NuPAGE LDS Sample Buffer (Invitrogen) and boiled for 10 min at 95 °C. Proteins were then resolved on NuPAGE 4–12% Bis–Tris gels (Invitrogen) and transferred to Nitrocellulose Membranes (0.2 µm, Thermo Scientific) using a Trans-Blot Turbo Transfer System (Bio-Rad). The membrane was then blocked by 5% (w/v) non-fat milk in PBST buffer, probed with Mouse Monoclonal anti-Flag M2 (Sigma-Aldrich) and Alexa Fluor Plus 800-conjugated goat anti-mouse IgG (Invitrogen) and visualized by fluorescence imaging. The membrane was subsequently incubated with horseradish peroxidase-conjugated anti-actin antibody, incubated with SuperSignal West Pico PLUS Chemiluminescent Substrate and then visualized by film developer.

### Statistics

The Student's $t$-test for two samples with unequal variances was used for statistical analysis in Microsoft Excel. $P < 0.05$ was considered as statistically significant.

### Reporting summary

Further information on research design is available in the Nature Portfolio Reporting Summary linked to this article.

## Data availability

All sequences are provided as Supplementary Information. The raw and analysed datasets generated during the study are available for research purposes from the corresponding authors on reasonable request. Source data are provided with this paper.

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

## Acknowledgements

We thank N. James, K. Ciazynska and J. Zurcher for providing Fluc, Spike and Cas9 sequences, respectively; A. Herrero Del Valle for providing the A549 cells; and M. Hoepfler for suggestions in RT–qPCR. V.R. discloses support for the research described in this study from the UK Medical Research Council (MC_U105184332) and the Wellcome Trust Senior Investigator award (WT096570). Y.D. discloses support for the research described in this study from the China Postdoctoral Science Foundation (PC2021083) and the Leverhulme Trust (ECF-2022-525). P.K.Z. discloses support for the research described in this study from the German National Academy of Sciences Leopoldina (LPDS 2021-14) and the EMBO (ALTF 778-2021).

## Author contributions

V.R. supervised the project. Y.D. and V.R. conceived the project. Y.D. performed most experiments and analysed the data. P.K.Z. ran the formaldehyde agarose gels, and cloned the V2 T2A-EGFP, T2A-Nluc and circZnf609 constructs. H.X. designed the RT–qPCR experiments and processed the resulting data. X.L. and Y.G. cloned the RCT constructs. X.L. performed western blotting analysis. Y.D. wrote the manuscript with revisions from P.K.Z., X.L. and V.R. and comments from all authors.

## Competing interests

Y.D. and V.R. are inventors on patent applications filed by the Medical Research Council based on this work, and co-founders of RNAvate Ltd.

## Additional information

**Extended data** is available for this paper at https://doi.org/10.1038/s41551-024-01306-3.

**Correspondence and requests for materials** should be addressed to Yifei Du or V. Ramakrishnan.

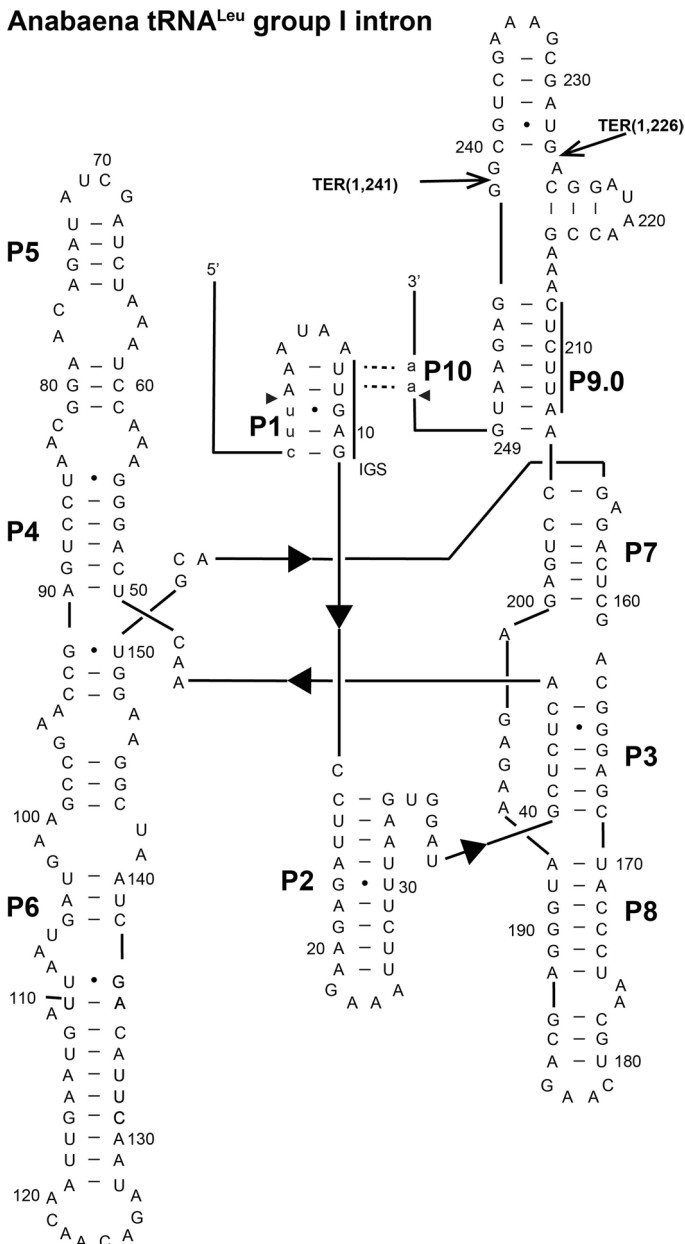

**Extended Data Fig. 1 | The Anabaena tRNA^Leu group I intron.** Secondary structure of the Anabaena tRNA^Leu group I intron. Anabaena intron consists of 10 base paired structures (P1-P10). The IGS forms P1 and P10 structures with flanking exons in proximity for splicing. TER: *trans*-excision ribozyme.

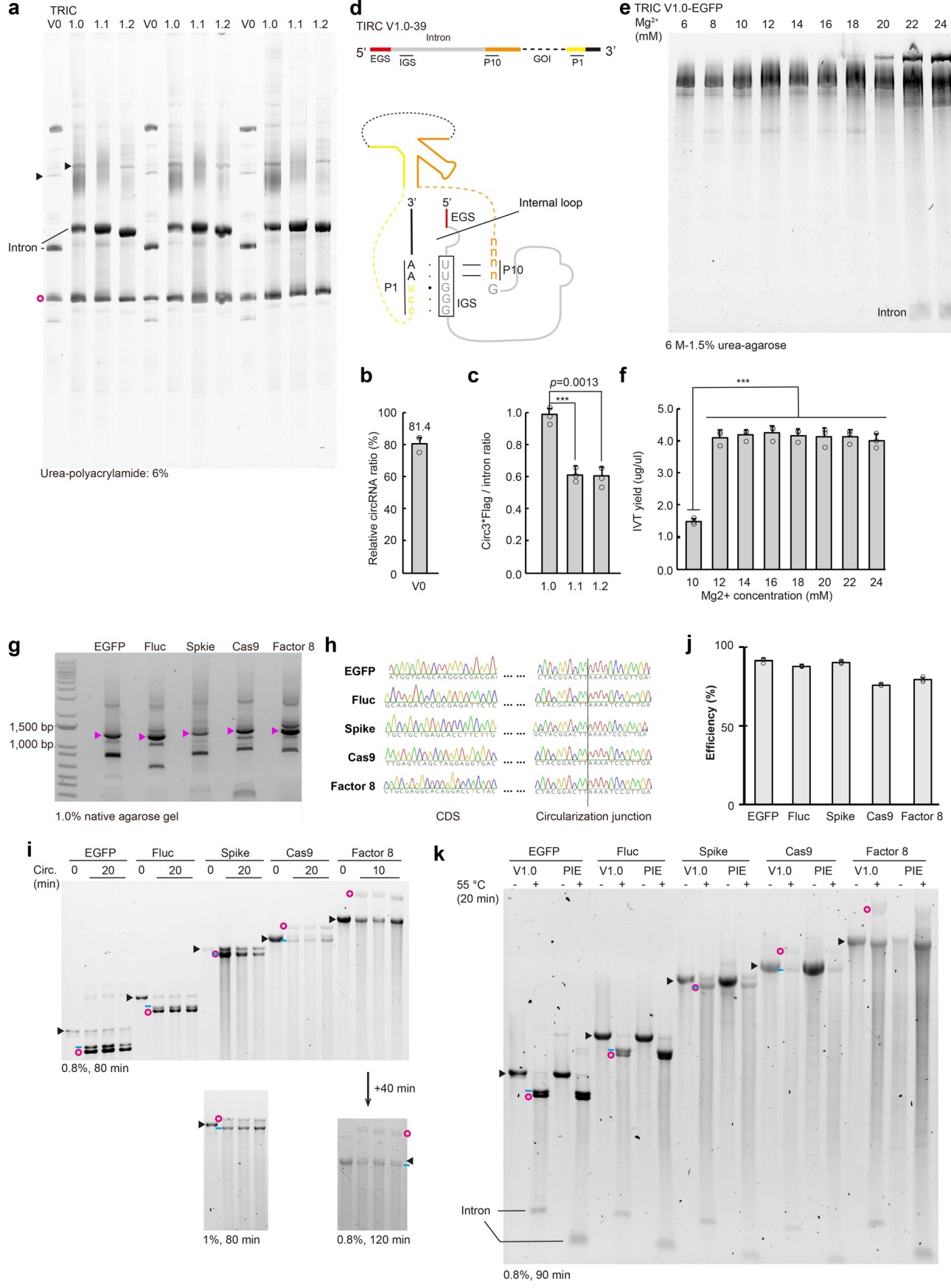

**Extended Data Fig. 2 | See next page for caption.**

**Extended Data Fig. 2 | Optimization of the TRIC-V1 for long circRNAs.**
(**a**) Co-transcriptionally circularized sample of V0, V1.0, V1.1, and V1.2 analysed on a 6% urea-polyacrylamide gel. (**b-c**) Each band in (**a**) were integrated by image J. The relative circRNA ratio in (**b**) is calculated by dividing the circRNA yield limit by the ratio of circRNA intensity to total intensity. The yield limit is defined as the ratio of the molecular weight of circRNA to that of FL. Circ3*Flag/intron ratio in (**c**) is given by intensity of circ3*Flag divided by intensity of intron. (**d**) Scheme of the V1. An EGS that base pairs with 3′ end of the construct was introduced. The introduction of EGS results in an internal loop that is essential for the ribozyme activity. (**e-f**) IVT were performed at $Mg^{2+}$ concentrations from 6 to 24 (**e**) and 10 to 24 (**f**) mM. Co-transcriptionally circularized was undetectable when $Mg^{2+}$ is ≤16 mM (e, urea-agarose gel) and IVT yield is significantly reduced when $Mg^{2+}$ is lower than 12 mM. IVT yield is given by ug of RNA produced per ul of IVT. (**g-h**) RT-PCR products of circularized EGFP, Fluc, Spike, Cas9, and Factor 8 were confirmed by agarose gel and Sanger sequencing. (**i, k**) FL samples were subject to 20 min of circularization (Circ.) (10 min for Factor 8) and analysed on native agarose gels. (**i**) The gel after imaging was extended for 40 min of running to separate FL and nicked RNA of Factor 8. The Spike samples were analysed on a 1% native agarose gel. (**k**) V1.0 exhibited similar efficiency (ratio of converted FL) as PIE. (**j**) Each band in (**i**) were integrated by image J. Circularization efficiency is the ratio between spliced FL and total FL. Data are mean ± SD for 3 independent replicates. ***p < 0.001, unpaired two-tailed t-test.

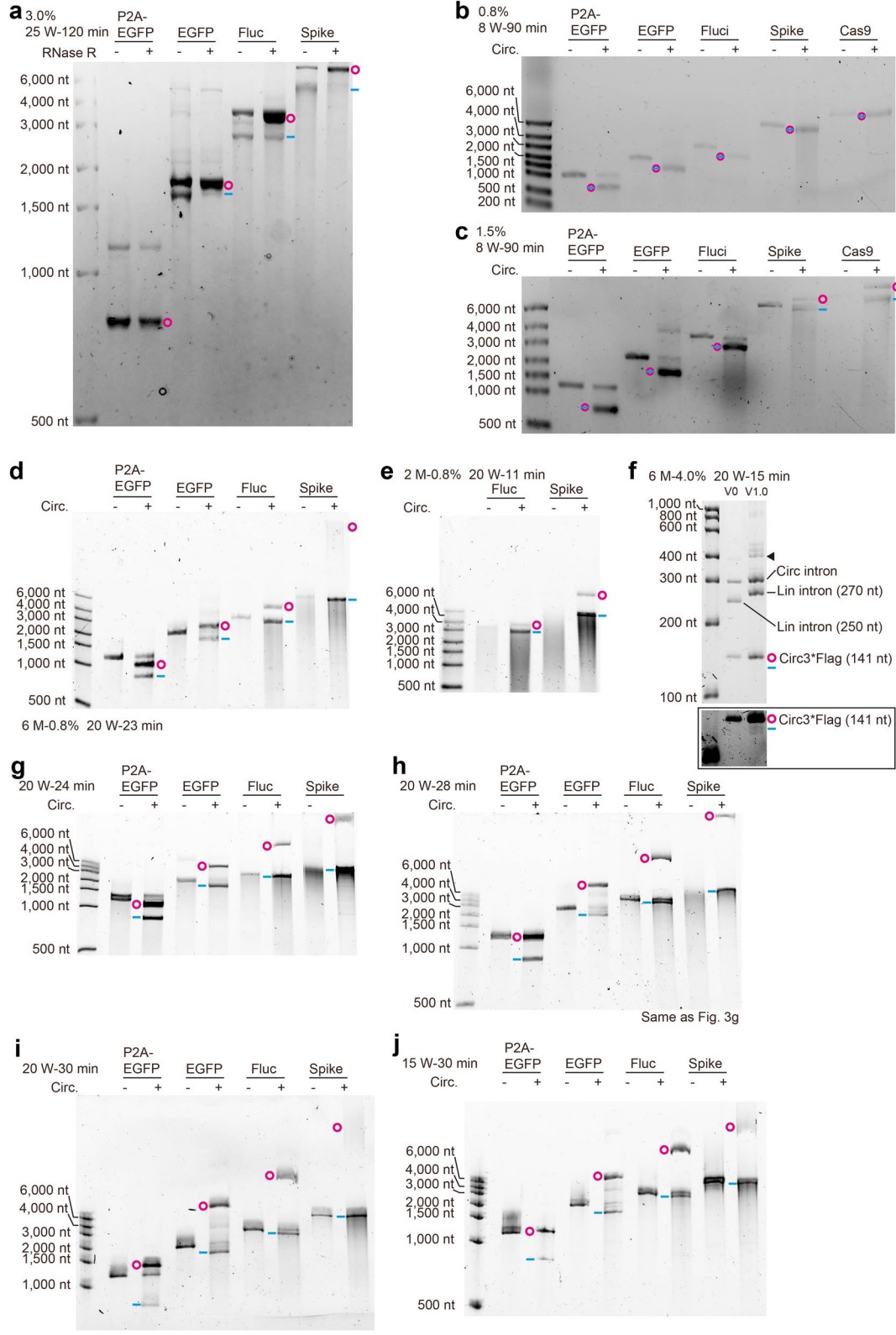

**Extended Data Fig. 3 | Analysis of circRNAs in agarose gels. (a)** Circularized V1.0 P2A-EGFP, EGFP, Fluc, Spike, and Cas9 were digested with RNase R and analysed in a 3% native agarose gel. **(b-c)** FL and circularized RNAs were analysed in 0.8% **(b)** and 1.5% **(c)** formaldehyde agarose gels operated in cold room. **(d-e)** FL and circularized RNAs were loaded onto 0.8% agarose gels with 6 M **(d)** or 2 M **(e)** urea. **(f)** Circularized V0 and V1.0 3*Flag were load onto a 6 M-4% urea-agarose gel. The insert shows the 141 nt nicked circ3*Flag. **(g-j)** Running power **(g-h)** and time **(i-j)** were explored for the 6 M-1.5% urea-agarose gels. Gel h is the same gel as the gel in Fig. 3g.

| | Running Settings | RNA length (nt) | | | | |
|---|---|---|---|---|---|---|
| | | < 500 | 500-1,000 | 1,000-3,000 | 3,000-5,000 | > 5,000 |
| Native agarose 0.8% | 25 W 70-120 min | − ○ | − ○ | − ○ | ✕ | − ○ |
| Native agarose 1.5% | | ✕ | ✕ | ✕ | − ○ | − ○ |
| Native agarose 3.0% | | ✕ | − ○ | − ○ | − ○ | − ○ |
| Urea-agarose 2 M-0.8% | 15-25 W 10-30 min | ✕ | ✕ | ✕ | − ○ | − ○ |
| Urea-agarose 2 M-1.5% | | ✕ | − ○ | − ○ | − ○ | − ○ |
| Urea-agarose 4 M-1.5% | | ✕ | − ○ | − ○ | − ○ | − ○ |
| Urea-agarose 6 M-1.5% | | − ○ | − ○ | − ○ | − ○ | − ○ |
| Urea-agarose 6 M-4.0% | | − ○ | − ○ | ✕ | ✕ | ✕ |
| Formaldehyde-agarose | 8 W /90 min | ✕ | ✕ | ✕ | ✕ | − ○ |
| Urea-PAGE | Hours | − ○ | ✕ | ✕ | ✕ | ✕ |

**Extended Data Fig. 4 | A summary for gel analysis of circRNAs.** The combination of agarose gels and urea- polyacrylamide gel enable separation of circRNAs (magenta circles) from their linear forms (blue lines) of various lengths. Red crosses suggest settings where separation is unachievable, or handling is difficult.

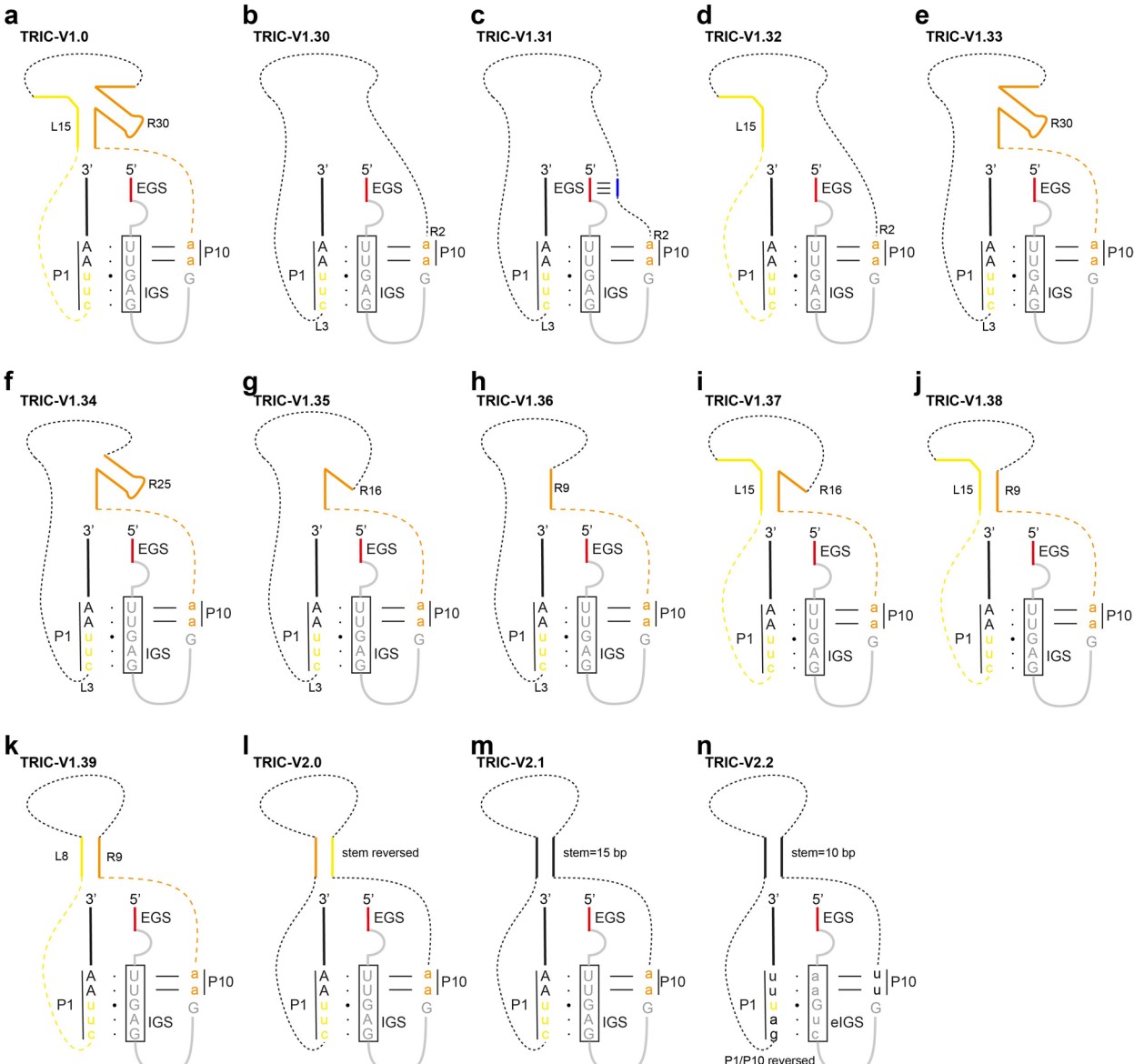

**Extended Data Fig. 5 | Diagram of V1 and V2 constructs tested in Fig. 4b.**
(**a-n**) Diagram of V1 and V2 variants. (**b-n**) The L15/R30 regions of the tRNA sequence in V1.0 were systematically engineered. In V1.30–V1.39, the tRNA sequence was gradually truncated. In V1.39, the L8/R9 region corresponds to the ACA sequence of the tRNA. For V2.0–V2.2, modifications to the anticodon stem were introduced: the sequence was reversed in V2.0, extended to 15 base pairs in V2.1, or the P1/P10 sequences were reversed in V2.2. Yellow and orange lines and letters represent native tRNA sequences.

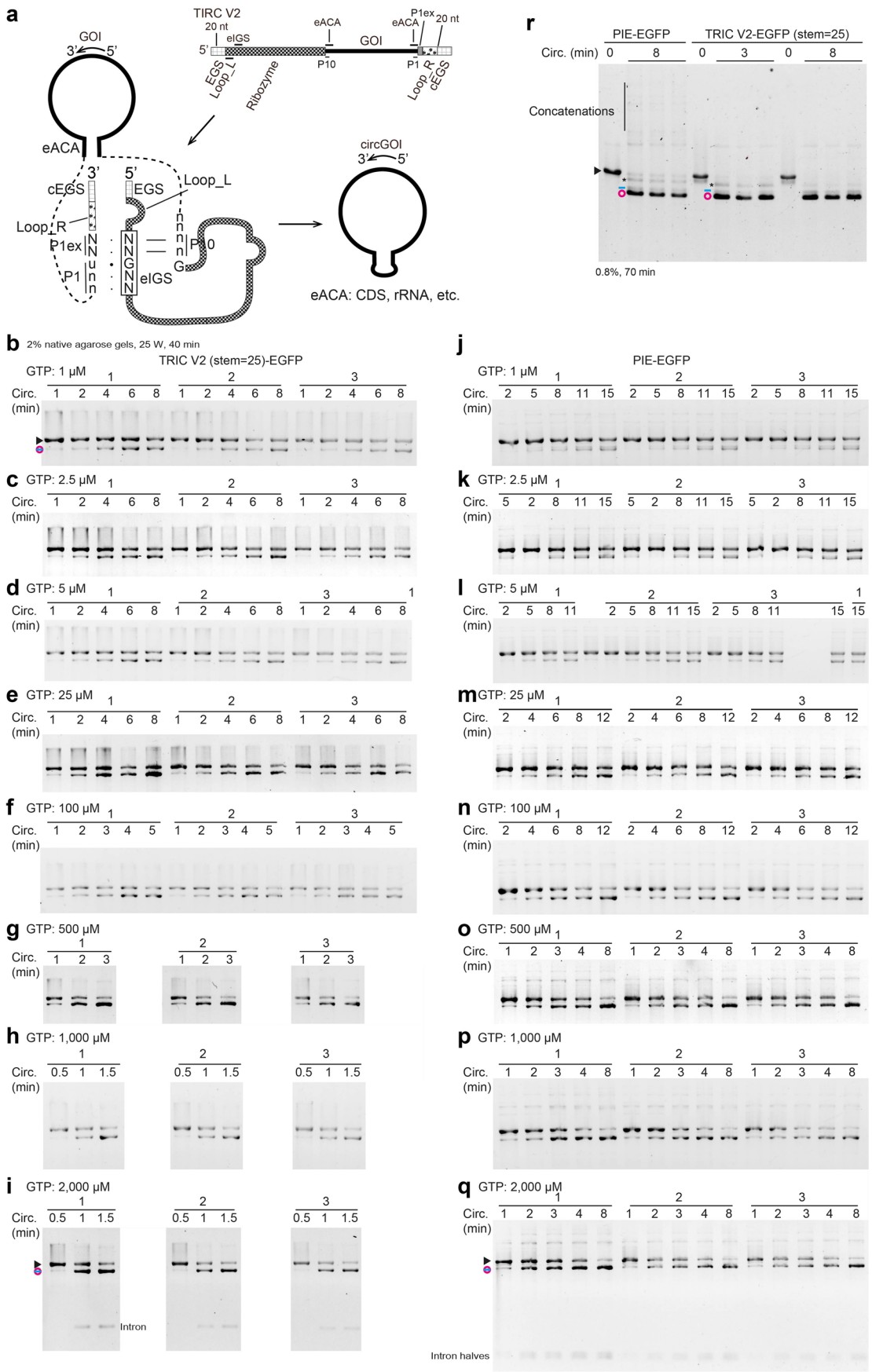

**Extended Data Fig. 6 | See next page for caption.**

**Extended Data Fig. 6 | Characterization of the TRIC-V2 (stem = 25) and the PIE.** (**a**) The design strategy of the TRIC-V2. cEGS: complementary sequence of the EGS. (**b-q**) Construct kinetics. Circularization of V2 (stem=25) EGFP (**b-i**) and PIE EGFP (**j-q**) were performed in 10 ul volumes at 55 °C for 0.5-15 min. 2 µl of 100 mM EDTA was used to stop circularization. GTP concentrations are 1, 2.5, 5, 25, 100, 500, 1,000, 2,000 µM. Samples were analysed on 2% native agarose gels. The Michaelis–Menten model was used to estimate the construct kinetics. (**r**) FL of V2 (stem=25) EGFP and PIE EGFP were circularized for either 3 and 8 min or 8 min and analysed on a native gel. Intensity of each RNA bands were integrated in image J and used for corresponding calculations.

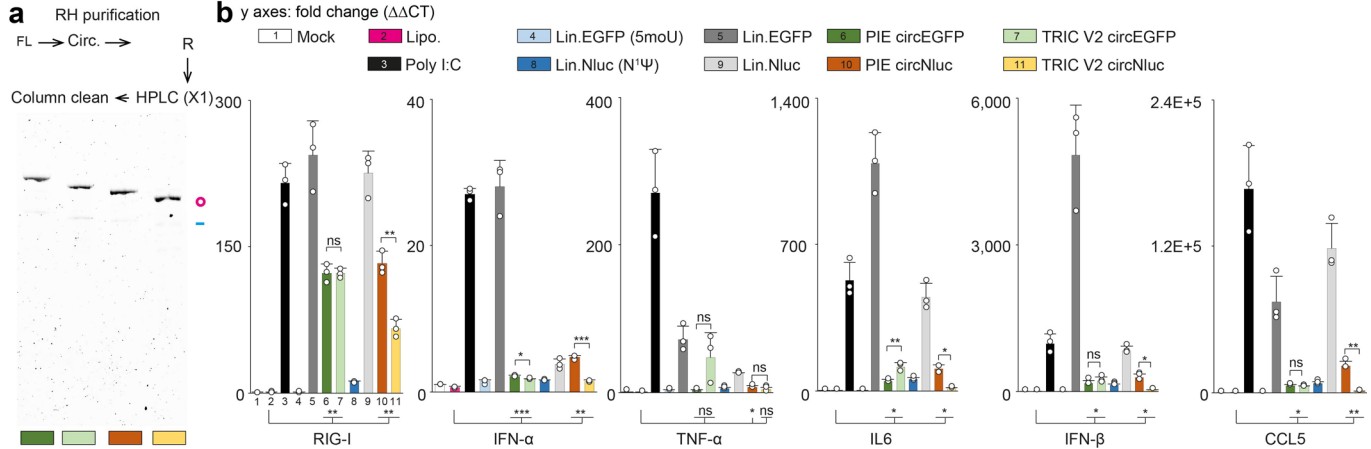

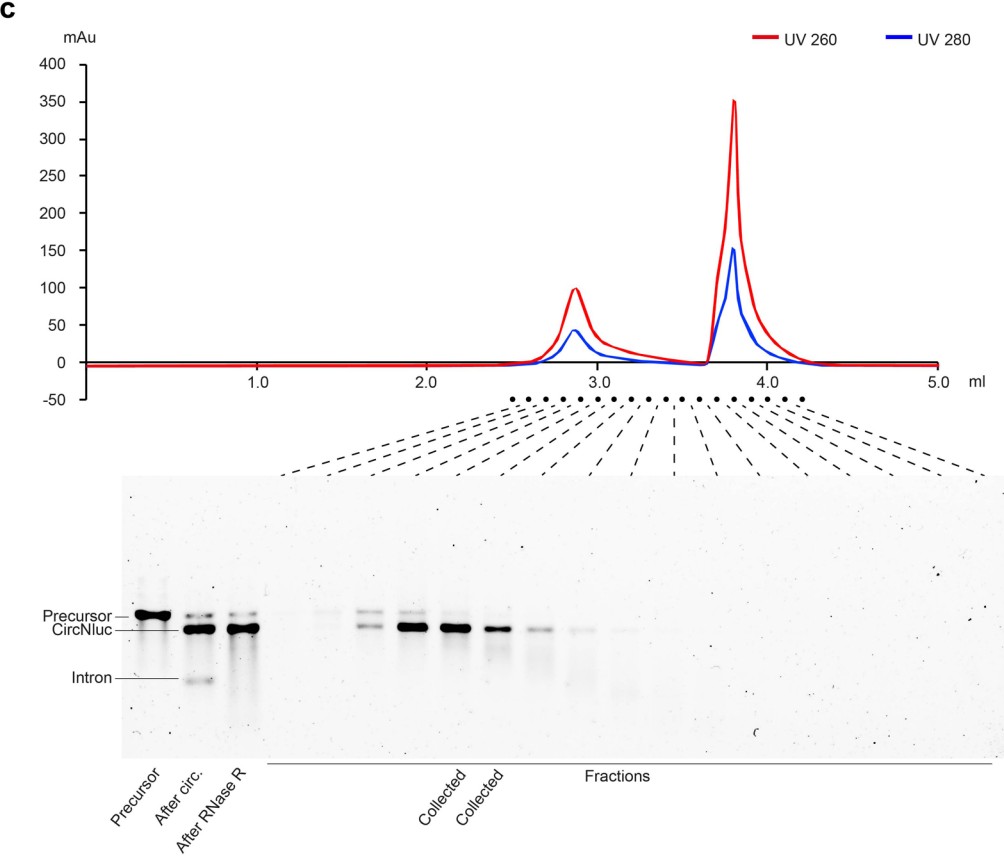

**Extended Data Fig. 7 | Immunogenicity of RH purified circRNAs. (a)** In the RH purification, circularized RNA was treated with RNase R and then purified by HPLC. The late circRNA fraction was column cleaned. 50 ng of each purified circRNAs was checked on a 6 M-1.5% urea-agarose gel. (**b**) 100 ng of each purified circRNAs, poly(I:C), and both modified and unmodified linear mRNAs (capped and poly-A tailed) was transfected into 10,000 A549 cells. mRNA levels of RIG-I

and several cytokines were monitored by RT-qPCR 24 h later. Y axes represent fold change compared to the Mock. Data are mean ± SD for 3 biological replicates. *p < 0.05, **p < 0.01 and ***p < 0.001, unpaired two-tailed t-test. (**c**) HPLC profile of the PRH purification. Fractions were analysed on a 0.8% native agarose gel (SYBR Green) run at 25 W for 30 min.

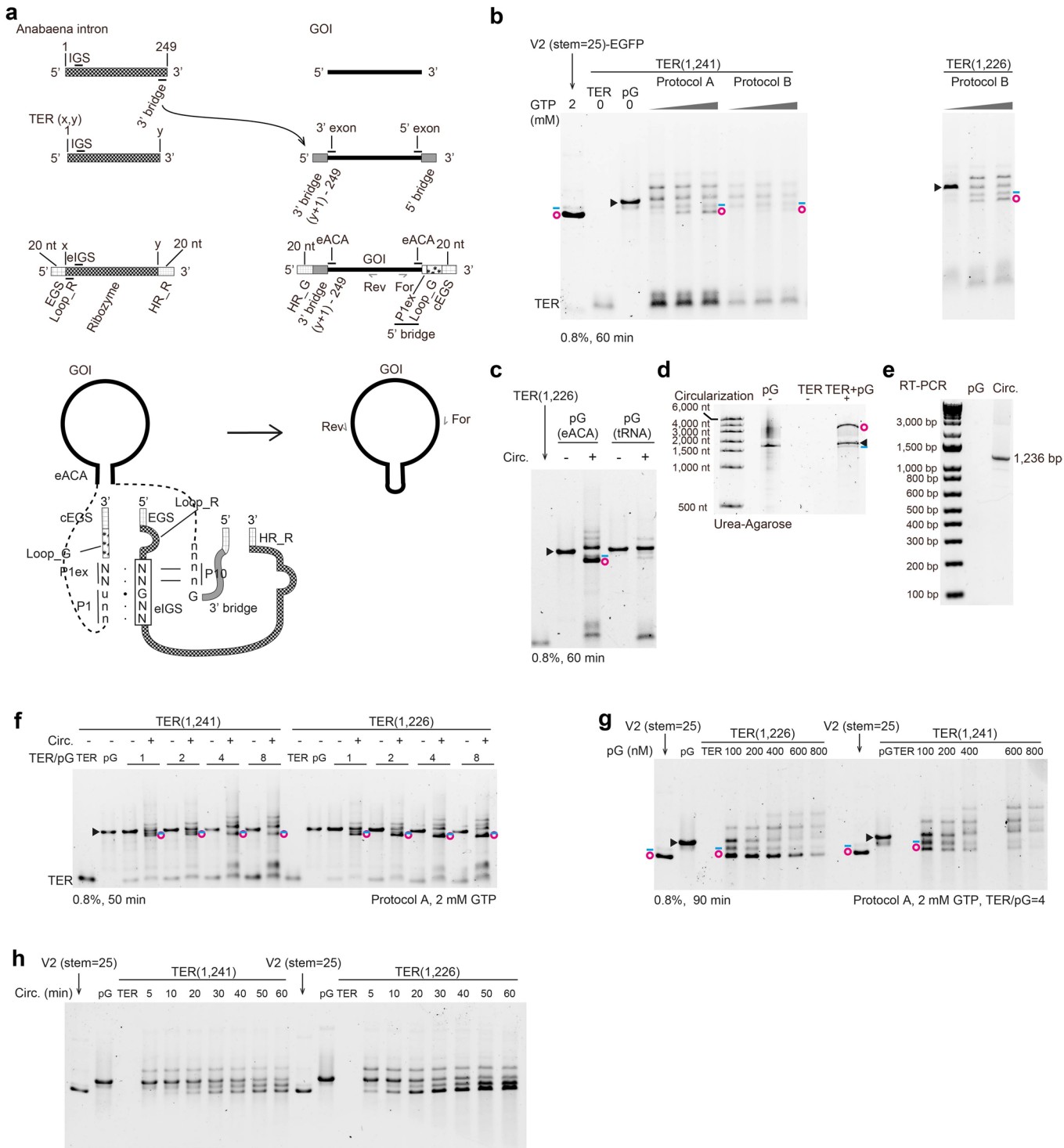

**Extended Data Fig. 8 | Synthesis of circRNAs by TERIC. (a)** The design strategy of the TERIC. **(b)** Circularization of EGFP using TERIC were conducted in 10 ul volumes with 0.1, 0.5, or 2 mM GTP. In protocol A, in vitro transcribed TERs and pGs were initially mixed at a 5:1 ratio, refolded, supplemented with splicing buffer, and heated at 55 °C for 20 min. In protocol B, TERs and pGs were individually refolded in splicing buffer, followed by mixing and heating at 55 °C for 20 min. The final concentration of pGs was maintained at 200 nM. Subsequently, splicing reactions were halted using 2 μL of 100 mM EDTA and loaded onto a 0.8% native agarose gel. Circularized V2 (stem=25) EGFP was loaded as a reference. **(c)** Circularization of EGFP using the TER(1,226). pGs either contains an eACA (stem=25) or tRNA sequence (stem=5). **(d)** TER(1,226), pG and circularized pG were analysed on a 6 M-1.5% urea-agarose gel. **(e)** RT-PCR using pG or circularized pG as template. Primers were indicated in (A). (F-H) TER to pG ratio **(f)**, concentration of pGs **(g)**, and duration for circularization **(h)** were optimized. 0.8% native agarose gel were used. The best reaction setting for TER(1,226) is 200-400 nM pG and 20-40 min of circularization in the presence of 2 mM GTP and 4 times amount of TERs vs pGs.

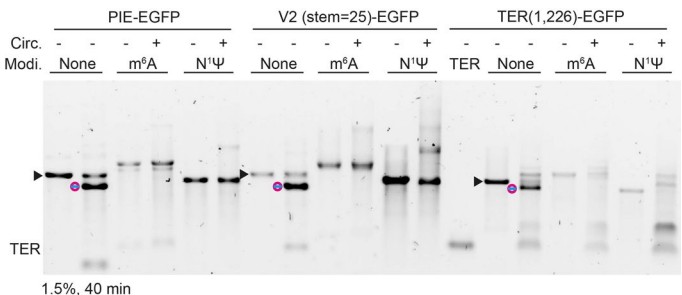

**Extended Data Fig. 9 | Synthesis of modified circRNAs.** Unmodified or m⁶A or N¹Ψ modified FL of PIE, V2, and pG of TERIC were subject to circularization and analysed on a 1.5% agarose gel. Black arrows indicate FL, magenta circles and blue lines indicate circRNA and its nicked product.

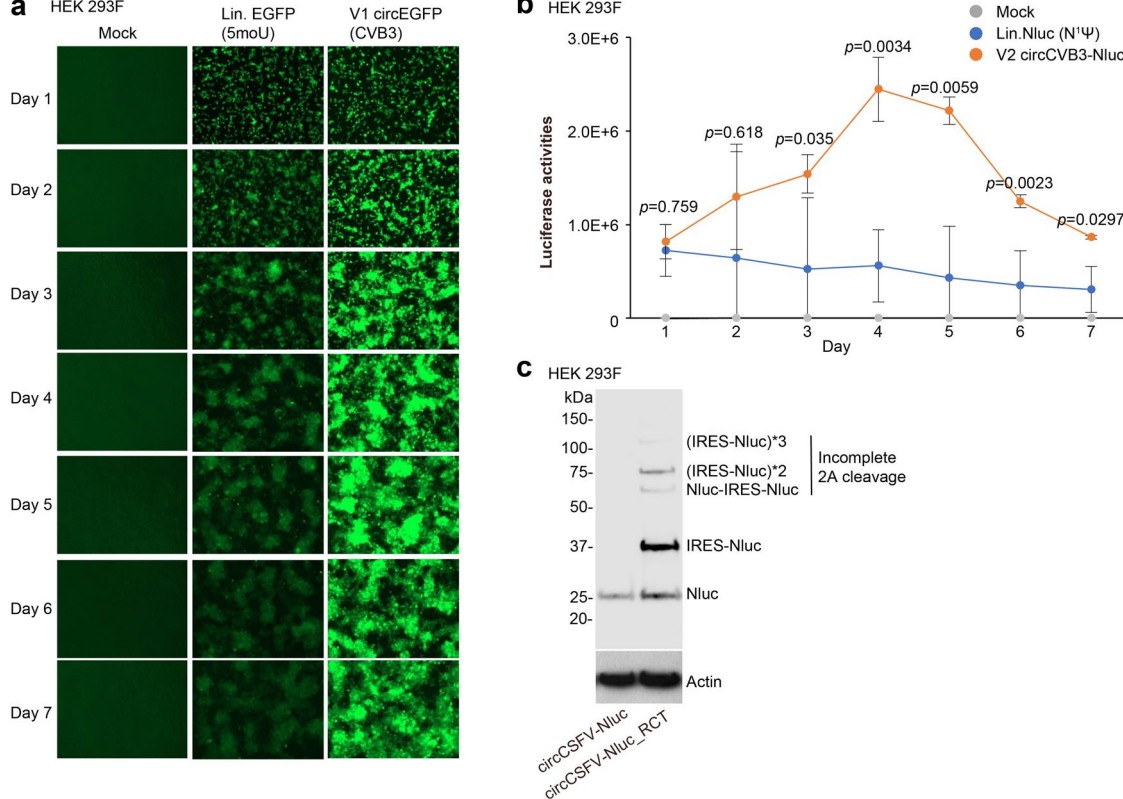

**Extended Data Fig. 10 | Protein expression from circRNAs in HEK 293 F cells.** (**a**) 0.04 pM of 5-methoxyuridine (5moU) modified linear EGFP and V1.0 circCVB3-EGFP were transfected into HEK 293 F cells. Expression of EGFP was monitored for 7 days. (**b**) 0.04 pM of N¹Ψ linear Nluc and V2 circNluc were transfected into HEK 293 F cells. Expression of Nluc was monitored for 7 days.

Data are mean ± SD for 4 biological replicates (3 replicates for Day 4 due to cell contamination). Unpaired two-tailed t-test. (**c**) Western blot analysis of Nluc expression from either single shot translation or RCT using CSFV IRES. Actin was used as internal reference. Polyproteins were observed.

# Reporting Summary

## Statistics

For all statistical analyses, confirm that the following items are present in the figure legend, table legend, main text, or Methods section.

| n/a | Confirmed | |
|---|---|---|
| ☐ | ☒ | The exact sample size (*n*) for each experimental group/condition, given as a discrete number and unit of measurement |
| ☐ | ☒ | A statement on whether measurements were taken from distinct samples or whether the same sample was measured repeatedly |
| ☐ | ☒ | The statistical test(s) used AND whether they are one- or two-sided *Only common tests should be described solely by name; describe more complex techniques in the Methods section.* |
| ☒ | ☐ | A description of all covariates tested |
| ☐ | ☒ | A description of any assumptions or corrections, such as tests of normality and adjustment for multiple comparisons |
| ☐ | ☒ | A full description of the statistical parameters including central tendency (e.g. means) or other basic estimates (e.g. regression coefficient) AND variation (e.g. standard deviation) or associated estimates of uncertainty (e.g. confidence intervals) |
| ☐ | ☒ | For null hypothesis testing, the test statistic (e.g. *F*, *t*, *r*) with confidence intervals, effect sizes, degrees of freedom and *P* value noted *Give P values as exact values whenever suitable.* |
| ☒ | ☐ | For Bayesian analysis, information on the choice of priors and Markov chain Monte Carlo settings |
| ☒ | ☐ | For hierarchical and complex designs, identification of the appropriate level for tests and full reporting of outcomes |
| ☒ | ☐ | Estimates of effect sizes (e.g. Cohen's *d*, Pearson's *r*), indicating how they were calculated |

*Our web collection on statistics for biologists contains articles on many of the points above.*

## Software and code

Policy information about availability of computer code

| Data collection | No code was generated in this research. Software used for data collection are: ViiA 7 Real-Time PCR System, Quantstudio Real-time PCR software v1.3, Bio-Rad ChemiDoc XRS+ Imaging System, EVOS FL microscope, Spark 10M plate reader. |
|---|---|
| Data analysis | Software used for data analysis are: Microsoft Excel, Image J |

For manuscripts utilizing custom algorithms or software that are central to the research but not yet described in published literature, software must be made available to editors and reviewers. We strongly encourage code deposition in a community repository (e.g. GitHub). See the Nature Portfolio guidelines for submitting code & software for further information.

## Data

Policy information about availability of data

All manuscripts must include a data availability statement. This statement should provide the following information, where applicable:
- Accession codes, unique identifiers, or web links for publicly available datasets
- A description of any restrictions on data availability
- For clinical datasets or third party data, please ensure that the statement adheres to our policy

All sequences are provided as Supplementary information. Source data for the figures are provided with this paper. The raw and analysed datasets generated during the study are available for research purposes from the corresponding authors on reasonable request.

# Research involving human participants, their data, or biological material

Policy information about studies with <u>human participants or human data</u>. See also policy information about <u>sex, gender (identity/presentation), and sexual orientation</u> and <u>race, ethnicity and racism</u>.

| | |
|---|---|
| Reporting on sex and gender | The study did not involve human research participants. |
| Reporting on race, ethnicity, or other socially relevant groupings | – |
| Population characteristics | – |
| Recruitment | – |
| Ethics oversight | – |

Note that full information on the approval of the study protocol must also be provided in the manuscript.

# Field-specific reporting

Please select the one below that is the best fit for your research. If you are not sure, read the appropriate sections before making your selection.

☒ Life sciences ☐ Behavioural & social sciences ☐ Ecological, evolutionary & environmental sciences

For a reference copy of the document with all sections, see nature.com/documents/nr-reporting-summary-flat.pdf

# Life sciences study design

All studies must disclose on these points even when the disclosure is negative.

| | |
|---|---|
| Sample size | 3-4 biological or independent replicates were used in all experiments to allow for statistical analysis. |
| Data exclusions | For the RT-qPCR experiments, 3 replicates were used for each biological sample. When the difference between the CT of a replicate and the CT of the remainder two replicates was larger than 1, this replicate was removed. Otherwise, no data were excluded from analysis. |
| Replication | All findings were replicated at least twice, once in the presented experiments and at least once in preliminary tests. |
| Randomization | Randomization was not used. |
| Blinding | Blinding was not needed. |

# Reporting for specific materials, systems and methods

We require information from authors about some types of materials, experimental systems and methods used in many studies. Here, indicate whether each material, system or method listed is relevant to your study. If you are not sure if a list item applies to your research, read the appropriate section before selecting a response.

### Materials & experimental systems

| n/a | Involved in the study |
|---|---|
| ☐ | ☒ Antibodies |
| ☐ | ☒ Eukaryotic cell lines |
| ☒ | ☐ Palaeontology and archaeology |
| ☒ | ☐ Animals and other organisms |
| ☒ | ☐ Clinical data |
| ☒ | ☐ Dual use research of concern |
| ☒ | ☐ Plants |

### Methods

| n/a | Involved in the study |
|---|---|
| ☒ | ☐ ChIP-seq |
| ☒ | ☐ Flow cytometry |
| ☒ | ☐ MRI-based neuroimaging |

# Antibodies

| | |
|---|---|
| Antibodies used | Monoclonal ANTI-FLAG® M2 antibody produced in mouse (F1804, Sigma-Aldrich); Goat anti-Mouse IgG (H+L) Highly Cross-Adsorbed Secondary Antibody, Alexa Fluor™ Plus 800 (A32730, Invitrogen); HRP Mouse monoclonal [mAbcam 8226] to beta Actin (ab20272, abcam). |

| Validation | The ANTI-FLAG M2 mouse, affinity purified monoclonal antibody binds to fusion proteins containing a FLAG peptide sequence.The antibody recognizes the FLAG peptide sequence at the N-terminus, Met-N-terminus,C-terminus, and internal sites of the fusion protein. The manufacturer shows a immunofluorescence result that indicates the validation of the product. https://www.sigmaaldrich.com/GB/en/product/sigma/f1804?utm_source=google&utm_medium=cpc&utm_campaign=15001183107&utm_content=127306761543&gclid=EAIaIQobChMIyrOr-JC7hQMVT5RQBh1CoAU6EAAYASAAEgIrcvD_BwE<br>HRP Mouse monoclonal [mAbcam 8226] to beta Actin, widely used as loading control, the manufacturer shows three western blot results that indicated the validation of the product. https://www.abcam.com/products/primary-antibodies/hrp-beta-actin-antibody-mabcam-8226-loading-control-ab20272.html |
|---|---|

## Eukaryotic cell lines

Policy information about cell lines and Sex and Gender in Research

| Cell line source(s) | HEK 293F cells were from Thermo Fisher, A549 cells were from Dr. Alba Herrero Del Valle. |
|---|---|
| Authentication | The cell lines were not authenticated. |
| Mycoplasma contamination | The cell lines were not tested for mycoplasma contamination. |
| Commonly misidentified lines<br>(See ICLAC register) | No commonly misidentified cell lines were used. |

