## [Peer Review File · Nature Biomedical Engineering]

Efficient circular RNA synthesis for potent rolling circle translation

Corresponding author: Yifei Du

Editorial note

This document includes relevant written communications between the manuscript's corresponding author and the editor and reviewers of the manuscript during peer review. It includes decision letters relaying any editorial points and peer-review reports, and the authors' replies to these (under 'Rebuttal' headings). The editorial decisions are signed by the manuscript's handling editor, yet the editorial team and ultimately the journal's Chief Editor share responsibility for all decisions.

Any relevant documents attached to the decision letters are referred to as **Appendix #**, and can be found appended to this document. Any information deemed confidential has been redacted or removed. Earlier versions of the manuscript are not published, yet the originally submitted version may be available as a preprint. Because of editorial edits and changes during peer review, the published title of the paper and the title mentioned in below correspondence may differ.

Correspondence

Thu 18 Jan 2024

Decision on Article nBME-23-2967

Dear Dr Du,

Thank you again for submitting to *Nature Biomedical Engineering* your manuscript, "Efficient circular RNA synthesis for potent rolling circle translation", and for your patience in waiting for the reviewer reports and our assessment of the feedback. The manuscript has been seen by four experts, whose reports you will find at the end of this message. You will see that the reviewers appreciate the work, and that they raise a number of fair technical criticisms that we hope you will be able to address.

When you are ready to resubmit your manuscript, please upload the revised files, a point-by-point rebuttal to the comments from all reviewers, the reporting summary, and a cover letter that explains the main improvements included in the revision and responds to any points highlighted in this decision.

Please follow the following recommendations:

- * Clearly highlight any amendments to the text and figures to help the reviewers and editors find and understand the changes (yet keep in mind that excessive marking can hinder readability).
- * If you and your co-authors disagree with a criticism, provide the arguments to the reviewer (optionally, indicate the relevant points in the cover letter).
- * If a criticism or suggestion is not addressed, please indicate so in the rebuttal to the reviewer comments and explain the reason(s).
- * Consider including responses to any criticisms raised by more than one reviewer at the beginning of the rebuttal, in a section addressed to all reviewers.
- * The rebuttal should include the reviewer comments in point-by-point format (please note that we provide all reviewers will the reports as they appear at the end of this message).* Provide the rebuttal to the reviewer comments and the cover letter as separate files.

We hope that you will be able to resubmit the manuscript within 12 weeks from the receipt of this message. If this is the case, you will be protected against potential scooping. Otherwise, we will be happy to consider a revised manuscript as long as the significance of the work is not compromised by work published elsewhere or accepted for publication at *Nature Biomedical Engineering*.

We hope that you will find the referee reports helpful when revising the work, which we look forward to receive. Please do not hesitate to contact me should you have any questions.

Best wishes,

Pep

—
Pep Pàmies
Chief Editor, Nature Biomedical Engineering

Reviewer #1 (Report for the authors (Required)):

The manuscript presents a novel trans splicing-based approach called Trans Ribozyme-based Circularization (TRIC), for the efficient synthesis of circular RNAs (circRNAs) that are over 8,000 nucleotides (nt) in length. The study also explores the use of viral Internal Ribosomal Entry Sites (IRESs) for Rolling Circle Translation (RCT) and assesses the immunogenicity and translation efficiency of the synthesized circRNAs. The TRIC method for circRNA synthesis is well-described, and the use of the Anabaena tRNA Leu group I intron as a trans ribozyme is justified. Clear figures and gel electrophoresis results convincingly support the efficiency of the TRIC approach. The optimization steps (e.g., introduction of an extended guide sequence and internal loops) are logically presented and supported by experimental data. The comparison of TRIC (V1.0) with the permuted intron-exon method (PIE) for circularizing various genes is informative. The rationale for suppressing co-transcriptional circularization to reduce RNA nicking is well-explained.

The experimental design was thorough, the study was well-executed, and the findings had potential implications for the advancement of mRNA therapeutics. The manuscript is generally well-structured. Thus, I believe the manuscript is suitable for publication pending minor revisions and clarification in certain sections.

Specific Comments:

1. The abstract provides a clear overview of the study, highlighting the main findings and implications. However, it was inappropriate to claim that circRNAs incorporated sequences from human 28S ribosomal RNA have low immunogenicity and are translated more efficiently than PIE circRNAs because PIE is a method to generate the circRNAs. The translation efficiency should be compared between the circular RNA with or without human 28S ribosomal RNA.
2. The abstract of the manuscript also requires clarification regarding the claim of a protein expression increase of more than 7,000-fold by utilizing viral internal ribosomal entry sites (IRESs) for the rolling circle translation (RCT). It should be explicitly stated that this increase is in comparison to a human OR4F17 IRES instead of the single-shot translation. Such a clarification will ensure accurate interpretation of the results and prevent readers from drawing inaccurate conclusions.
3. In page 4 of the manuscript and as depicted in Figure 3, the length of P2A-EGFP is 813 nt, while FL EGFP measures 1963 nt. Please explicitly delineate the different modules of FL EGFP resulted in longer sequence compared to P2A-EGFP in the manuscript.
4. The exploration of the modified circRNAs using TERIC approach is innovative. The results indicating the synthesis of modified circZnf609 are compelling. However, circZnf609, being an endogenous molecule, inherently lacks immunogenicity. Please discuss the rationale for choosing circZnf609 instead of virus circSpike for testing immunogenicity.
5. The investigation of translation efficiency and the use of viral IRESs for RCT are interesting. The results,

especially the significant increase in translation efficiency with CSFV IRES-RCT, are noteworthy. However, in Fig. 7f, the expression level of V2-CVB3-Nluc is higher than those of N1 pseudouridine-modified Nluc and CSFV-Nluc-RCT. It will be very interesting if the author can show the RCT of V2 circCVB3-Nluc also can get more than 10-fold enhancement than V2 circCVB3-Nluc only.

6. During the RCT, as the authors described that the stop codons have been eliminated, the IRES sequence is also translated, leading to the synthesis of an IRES encoded additional peptide. Does the N-terminus of the POI carry at least a portion or the entire IRES translated peptide sequence after P2A cleavage? This is my main concern, and I hope the authors could provide a Western blot to show the protein molecular weight to confirm it contains the additional IRES peptide or not.

Reviewer #2 (Report for the authors (Required)):

Du et al. presented Anabaena group I intron-based trans ribozyme approach for circular RNA generation (TRIC approach). They showed that the method is independent of bacterial sequences, outperforms PIE method that is currently the most utilized method for RNA circularization, and allows for incorporation of RNA modification. Moreover, they proposed rolling circle translation (RCT)-based enhancement of GOI expression by utilizing and modifying viral IRES.

There are debating concerns in terms of potential innate immunity induction in the bacterial sequences remaining in the circular RNA generated by PIE method. To overcome the issues, several papers recently described already group I intron-based trans ribozyme approach to generate circular RNA without undesired intronic sequences (NAR 51, e78 (2023), Mol Ther Nucleic Acids 33, 587-598 (2023): references 40 and 41 in this manuscript). Main difference of the present study from the previous ones is that the authors here engineered, modified, and optimized the composition of Anabaena tRNA^{Leu} group I intron, in contrast that previous papers engineered Tetrahymena group I intron. Therefore, the concept of the trans ribozyme approach was already suggested by other groups. Moreover, group II-ribozyme based PIE method was also suggested as the circularization approach including the desired GOI only (bioRxiv, 2022, doi: <https://doi.org/10.1101/2022.05.31.494115>). Although the authors proposed the RNA circularization method superior to the current PIE method and efficient RCT method capable of highly improving GOI expression, following concerns and lack of clear data presentation should be addressed.

1. Line 42 and 208-209: "its inability to synthesize modified circRNAs", "The PIE, and likely the TRIC as well, are unable to synthesize modified circRNAs since modifications could disrupt the ribozyme activity.

 That's not true. 5% m6A-modified circRNA was reported to be generated using PIE method (Nat. Biotechnol. 2023, 41, 262-272: Reference 7 in the manuscript). Certain % of modified nucleotide incorporation condition does not affect ribozyme folding and its activity.

2. Line 45-46: "Urea-polyacrylamide gels effectively distinguish circRNAs from their linear forms but are limited to <500 nt."

 Longer circRNAs (~1000-nt) also were reported to be efficiently distinguished from their linear forms in urea-polyacrylamide gel (Mol Ther Nucleic Acids 2023, 33, 587-598: Reference 41 in the manuscript).

3. Line 91-92: if the concentration of Mg²⁺ and NTP were lowered to avoid nicking during IVT, transcription yield would be also lower. How much transcription yield was affected by lower concentrations of Mg²⁺ and NTP? Is it okay for large scale production of circRNA?

4. Circularization junctions should be confirmed for each longer GOI (e.g. Spike, Cas9, Factor 8).

5. In figure 2i, why circRNA bands are much thicker than 0 min precursor RNA bands? Same amounts of RNAs should be loaded to compare precisely.

6. In figure 3g, h, i, why Spike's nicked circRNA band is much thicker than precursor RNA band? And the RNA amounts in lanes Circ(-) and Circ(+) in each GOI are different. Same amounts of RNAs should be loaded to compare precisely.

7. In figure 5c, why the bands of precursor RNAs and circRNAs are much thicker than 0 min precursor RNA bands? Same amounts of RNAs should be loaded to compare precisely.

8. In figure 5d, why precursor RNAs are much thicker than 0 min precursor RNA bands? Same amounts of RNAs should be loaded to compare precisely.

9. TRIC and PIE are based on the same ribozyme, Anabaena group I intron, and use two consecutive transesterification reactions for the splicing. In line 162-165, is there any possible reason why the ribozyme kinetics of TRIC is better than that of PIE? Any previous studies and/or experimental evidences to support the proposed mechanism for better kinetics of TRIC over PIE should be provided.

10. In line 166-167, concatenations seemed to be observed for TRIC V2 also, even though less than PIE. Is this because of post-transcriptional circularization using "high concentrations of precursor RNAs", as observed for chemical or ligase based method using precursor RNAs? The generation of concatenations would cause inefficient large scale production of circRNA. Then, post-transcriptional circularization proposed in this study would be also impractical for large scale circRNA production, compared to current co-transcriptional method. Concatenations observed for post-transcriptional circularization using TRIC V2 should be more reduced.

11. In Figure 5g, what is the light salmon-colored column? Nicking ratio?

12. Is AKTA pure micro system FLPC or HPLC? Chromatograms and purification condition to check circRNA peak among various RNAs and the efficiency during purification should be presented.

13. In line 262-264, "Recent studies have reported *Tetrahymena thermophila* group I intron-derived constructs for clean RNA circularization. However, these constructs resemble the V1.30, exhibiting lower efficiency compared to V1.0 and PIE.

 The strategy in Reference 40 & 41 utilized different kind of group I intron (*Tetrahymena* group I intron) from V1.30, so how can you expect it to be less efficient? On what basis are the structures from *Tetrahymena* group I intron similar to the V1.3 structure? In the Reference 40 and 41, the circularization efficiency was observed to be comparable to or more than PIE. Direct experimental comparison between different constructs should be performed to conclude.

14. In line 343-345, is it possible that 55oC 20 min heating increases nicking and degradation of RNAs? The condition with 10 mM MgCl₂ and heating for post-transcriptional method would be favorable condition for nonspecific nicking and hydrolysis.

15. In Fig.7, the authors showed higher translation efficacy of TRIC circRNAs, compared to PIE circRNAs. If the expression efficacy of TRIC circRNAs is really higher than PIE circRNAs, what is the reason for higher expression? Comparison of innate immunity data between TRIC and PIE seems inconclusive. To compare the translation efficacy more precisely, the exact amount of circRNAs present in the transfected cells should be also compared.

16. In the rolling circle translation (RCT) approach, isn't it possible that peptide fragment downstream P2A site of the GOI may be generated via readthrough translation of modified IRES? If so, the unwanted products may cause serious concerns to the cells/tissue administered the circRNAs that can perform RCT. How you can remove the unwanted expression products from RCT circRNA?

17. Several previous studies proposed efficient circRNA generation without any extraneous sequences caused by PIE method. To show more advanced improvements and provide practical implication using the method developed in this study, in vivo expression data showing more durable and higher expression of GOI should be provided and compared.

Reviewer #3 (Report for the authors (Required)):

Du et al. report a novel strategy for the in vitro production of large circular RNAs based on the action of group I intron trans-splicing. In particular, the *Anabaena* tRNA LEU group I intron was used and engineered for RNA circularisation. A prototype trans acting ribozyme circularisation (TRIC) system was successively optimised, including ribozyme structure and circularisation conditions, to show significantly better performance than the corresponding circularisation by the PIE method. Importantly, the developed TRIC construct has minimal sequence requirements and the circularised RNA product contains only minimal portions of the native splicing sequence. The technique is demonstrated for the efficient circularisation of RNAs between 1000 and 9000 nt in length. Furthermore, the authors have developed a convincing protocol

for the analysis and purification of the circularised product based on denaturing agarose gels. Immunogenicity studies were carried out, including the adaptation of TRIC to circularise modified RNA using a trans excision ribozyme strategy. Finally, the authors demonstrate the efficient generation of RCT constructs using their novel technique, which are translated with high efficiency.

This work represents a technical advance with implications for the field of circRNA synthesis and application in molecular biology and medicine. It is a very timely contribution, as the efficient and precise production of circRNAs is of paramount importance for their use in RNA therapies. In addition, the reported analytical developments are relevant to the broader field of large RNA preparation and analysis.

Specific comments

1. The authors should also consider the TORNADO system in their introduction and discussion (Litke & Jaffrey, Nat Biotechnol. 2019; 37(6): 667-75) It was developed for RNA circularisation in cells, but can also be used for RNA circularisation in vitro.
2. The paper is less clear on terminology in some places. For example, when the term circ3*Flag is used for the first time, it would be helpful to explain that this is the desired circularised product. The same applies to ACA. Figure 2a shows that this is the anticodon arm. However, the abbreviation should also be defined in the text when it is used for the first time. There may be other examples of this that require attention.
3. Page 3, line 72/73: The authors say: "Due to the minimal FL, we concluded that the TRIC-V0 construct circularizes the 3*Flag sequence efficiently during IVT." This statement needs to be supported by quantitative data. Several byproducts are seen and by visual inspection the amount of remaining FL is not really minimal.
4. Page 3, line 76/77: "introducing a 20 or 23 bp EGS and internal loops to TRIC-V0 (Fig.2f)". There is only one internal loop seen in the structures in Fig. 2f. What is meant is probably "differently structured internal loops"? Please clarify.
5. Page 4, line78: "SV1.0 yielded the highest circRNA to intron ratio". What is the ratio of this variant compared to all other variants? Quantitative data should be provided.
6. Page 4, line 83-84: "Gel analysis revealed that V1.0 has a higher co-transcriptional splicing efficiency than the PIE for EGFP,.....and Cas9 (Fig. 2h)." This is an invalid statement as the gel only shows the disappearance of FL and not the appearance of defined splicing products. Although the consumption of FL is likely to be a result of splicing, it cannot be concluded as stated.
7. Page 5, line 142 – 144: How is V2 derived from V1? A sentence or two should be added to make this clear.
8. References: There are a number of references where there is a lack of information. (e.g. 7, 20, 27, 35, 36, 37).
9. Figure 2h: The vertical black lines indicating splicing products are barely visible and should be highlighted.
10. Figure 4: The structural differences between the variants are not well illustrated. A figure showing which parts of TRIC are varied from V1.30-39 and from V1 to V2 would be desirable. What is the structural difference between V1.30 and V1.31? According to Figure 4b, both have L/R: 3/2, no stem, P1/P10?
11. Figure 5g: The illustration lacks the meaning of the orange bar. I learned from the caption that it refers to nicking.

Reviewer #4 (Report for the authors (Required)):

This manuscript presents a novel trans Anabaena tRNA^{Leu} group I intron mediated methods for synthesis of circRNAs which outperform the more general used permuted intron-exon method (PIE) system. The authors demonstrate that the resulting circRNA has low immunogenicity and able to circularize very large circRNA (up to 8000 nts). It is also demonstrated that if stop codons are removed, the translation can efficiently read through highly structured IRESs, enhancing the translation efficiency by multiple rounds of initiation.

The novelty of the method itself is lowered somewhat in the light of the recent method papers (ref 40 and 41) using tetrahymena self-splicing intron, which also is non-permuted and scarless. The authors make comparison to these and find their method only to be slightly better.

In general, the manuscript is well written and covers a lot of details that are of interest to scientists who want to synthesize and purify circRNA and use it as templates for translation. circRNA has recently attracted a lot of attention and this manuscript will be interesting for a broader audience. However, there are some critical points that must be addressed before publication:

A general problem in the data analysis is that the disappearance of precursor is used as a measure of how efficient the splicing reaction is. But the disappearance of something is a poor measure for the yield because degradation of the RNA is a significant factor. Rather, the amount of circRNA that is produced should be used. In one instance the authors use the circRNA for quantification, but the methods for calculations are very poorly described in the methods section.

Fig. 1: Please indicate with arrows the 1. and 2. step attacks in b, c and d as in a)

Fig. 2b: It is a bit unclear how much belongs to panel a and b.

Fig. 2f: Why is there so much more linear intron in v. 1.1 and 1.2 when there is not an increase in circRNA and there is no precursor left?

Fig. 2h: It is not possible to conclude anything regarding splicing efficiency from this gel (As it is done in line 83-84).

Section about separation on agarose gel (from line 101). The separation of circRNA from precursor is very dependent on the circularization method, so to conclude that it is possible is too general. E.g. for T4 ligation, where the precursor and circle are of same length, separation may not be possible with these gels. Furthermore, it has been shown that EX-gels from Thermo Fisher separate circRNA and linRNA efficiently and quickly, which should be mentioned in the discussion.

Extended data figure 3b-c: What are the extra bands at the other samples (not Spike and Cas9)? Should comment on this.

Extended data figure 3h: What does "Same as Fig.3g" refer to?

Fig. 4: How are the efficiencies in this figure quantified? And how long is each circle? The length of the circle of course needs to be considered when quantifying.

Extended data Fig.4. There is not legend to explain all the symbols. Maybe this should be a Table rather than a figure?

Fig. 5: Why use gels in this figure that does not separate the circles from nicked RNA? Would make a lot more sense to run it on gels that separate.

Fig. 5e: Would the Michaelis-Menten analysis give the same result if you quantify the amount of circRNA produced rather than the amount of precursor that disappears? It is a rather indirect way of quantifying it.

Fig. 5f and g: The annotation of the "Nicked" sample is missing in Fig. 5g. Also, was the gel in Fig. 5f used to make the quantifications in 5g? How was it possible to separate the quantification of only the circRNA or the nicked RNA? These bands are so close that there would be intensity bleeding over. Furthermore, what do the 50% and 20% marks in 5g indicate? And why is the figure cut at the top part, so the V2-EGFP at 8 min cannot be seen?

Extended figure 6: If this is RH treatment, should the R and HPLC (X1) be switched around in a?

Extended figure 7B: Typo: "Ptotocol".

Figure 6h: Would have been nice to see the "none" lanes for TERIC as well.

The whole RCT section (and in particular fig. 7f): No comments are made about the CVB3 data in this figure. Also, it is misleading to say that "Translation from RCT is increased by over 7,000-fold" (headline), since it is

not related to the linear mRNA or CVB3 data at all. So, the comparison does not make sense. In fact, the data that you compare, and which is 7000-fold increased, is only a little more than 50% compared to the linear mRNA. So, this point is clearly overstated and should be moderated – also in the abstract.

Fig. 7: How was the Luc data measured . The errors in panel a unrealistically low – is that technical replicates?

Extended data Fig. 8: Need more explanation.

Methods: In the first paragraph of “Analysis of circularization efficiency, yield, nicking, and ribozyme kinetics”, the quantification technique is very poorly described. If FL means “Full length precursor”, as stated in the beginning, the “spliced FL” does not make sense. Furthermore, what does “proportionally calculated” mean? This section needs clarity to enable the reader to reproduce the quantification.

Fri 23 Aug 2024

Decision on Article NBME-23-2967A

Dear Dr Du,

Thank you for your patience in waiting for the feedback on your revised manuscript, "Efficient circular RNA synthesis for potent rolling circle translation". Having consulted with the original reviewers (whose comments you will find at the end of this message; however, Reviewer #4 unfortunately hasn't provided a report), I am pleased to write that we shall be happy to publish the manuscript in *Nature Biomedical Engineering*, provided that the points specified in the attached instructions file are addressed.

When you are ready to submit the final version of your manuscript, please upload the files specified in the instructions file.

Also, please address the minor points raised by Reviewers #2 and #3, and provide a point-by-point reply.

We encourage authors to take up transparent peer review. If you are eligible and opt in to transparent peer review, we will publish, as a single supplementary file, all the reviewer comments for all the versions of the manuscript, your rebuttal letters, and the editorial decision letters. **If you opt in to transparent peer review, in the attached file please tick the box 'I wish to participate in transparent peer review'; if you prefer not to, please tick 'I do NOT wish to participate in transparent peer review'**. In the interest of confidentiality, we allow redactions to the rebuttal letters and to the reviewer comments. If you are concerned about the release of confidential data, please indicate what specific information you would like to have removed; we cannot incorporate redactions for any other reasons. More information on transparent peer review is available.

Best wishes,

Pep

Pep Pàmies
Chief Editor, Nature Biomedical Engineering

P.S. Nature Portfolio journals encourage authors to share their step-by-step experimental protocols on a protocol-sharing platform of their choice. Nature Portfolio's Protocol Exchange is a free-to-use and open resource for protocols; protocols deposited in Protocol Exchange are citable and can be linked from the published article. More details can be found at www.nature.com/protocolexchange/about.

Reviewer #1 (Report for the authors (Required)):

In this revised manuscript, Yifei Du and coworkers have addressed all my concerns and other reviewers from the prior submission. The manuscript has been improved significantly as well as the incorporation of new comparison with other methods. Of particular relevance to my comments, they have now demonstrated IRES-derived translation products by western blots, this was my principal sticking point with the earlier submission, and I thus now recommend publication in *Nature Biomedical Engineering*.

Reviewer #2 (Report for the authors (Required)):

See attached document. **Appendix #1**

Reviewer #3 (Report for the authors (Required)):

The authors have adequately addressed the reviewers' comments and made the appropriate improvements. Suggested additional experiments were performed, missing data were added, and the manuscript was revised to correct typographical errors, add missing details, and include new figures. The overall quality and clarity of the manuscript has improved. My personal comments are well considered.

However, there is still one minor technical concern that I would like to mention:

The authors use the disappearance of the FL as an indicator of circularization. In the text, line 73/74, they still say: "Due to the minimal remaining FL, we concluded that the TRIC-V0 construct circularizes the 3*Flag sequence efficiently during IVT (Extended Data Fig. 2 a-b)." Based on the data in the extended data Fig. 2a, the relative FL ratio was calculated to be 3.6. I don't think one can say EFFICIENT circularization just based on the disappearance of FL. The authors should find a way to calculate or at least estimate the amount of circularized product (seen in the gel in Fig. 2a) relative to FL and the other splicing products to have a better measure of circularization efficiency.

Nature Biomedical Engineering is a Transformative Journal. Authors may publish their research with us through the traditional subscription access route, or make their paper immediately open access through payment of an article-processing charge. More information about publication options is available.

You may need to take specific actions to comply with funder and institutional open-access mandates. If the work described in the accepted manuscript is supported by a funder that requires immediate open access (as outlined, for example, by Plan S) and your manuscript was originally submitted on or after January 1st 2021, then you will need to select the gold OA route. Authors selecting subscription publication will need to accept our standard licensing terms (including our self-archiving policies), and these will supersede any other terms that the author or any third party may assert apply to any version of the manuscript.

Appendix 1

Reviewer #2:

Authors properly responded to most of raised issues except several points as follows.

1. Line 42 and 208-209: “its inability to synthesize modified circRNAs”, “The PIE, and likely the TRIC as well, are unable to synthesize modified circRNAs since modifications could disrupt the ribozyme activity.

→ That’s not true. 5% m6A-modified circRNA was reported to be generated using PIE method (Nat. Biotechnol. 2023, 41, 262-272: Reference 7 in the manuscript). Certain % of modified nucleotide incorporation condition does not affect ribozyme folding and its activity.

Response: We meant 100% modified, as can be seen in paragraph 2 of section “Synthesis and immunogenicity of modified circRNAs”. To clarify this point, ‘fully’ has been added before ‘modified circRNAs’.

→ **Comment: Properly answered.**

2. Line 45-46: “Urea-polyacrylamide gels effectively distinguish circRNAs from their linear forms but are limited to <500 nt”

→ Longer cRNAs (~1000-nt) also were reported to be efficiently distinguished from their linear forms in urea-polyacrylamide gel (Mol Ther Nucleic Acids 2023, 33, 587-598: Reference 41 in the manuscript).

Response: We agree that circRNAs >500 nt may also be separated by urea-PAGE if required. Our rationale for suggesting urea-PAGE is limited to 500 nt (see reference 18, page 225, note 5) is that analysing long circRNAs in urea-PAGE requires low percentages of acrylamide (4% or lower), which increases gel handling difficulties, and long running times. Both of these make analysis of long circRNAs (>500 nt) by urea-PAGE impractical.

→ **Comment: Properly answered.**

3. Line 91-92: if the concentration of Mg²⁺ and NTP were lowered to avoid nicking during IVT, transcription yield would be also lower. How much transcription yield was affected by lower concentrations of Mg²⁺ and NTP? Is it okay for large scale production of circRNA?

Response: We thank the reviewer for making this important point. We measured the total RNA yield under our IVT conditions (6 h) using Mg²⁺ concentrations between 10 and 24 mM and found that the yield remained unchanged when Mg²⁺ is higher than 12 mM (Extended Data Fig. 2f). As a result, using Mg²⁺ concentrations between 12-16 mM would lead to higher circRNA yield compared to 18 – 24 mM, as co-transcriptional circRNA nicking is reduced.

→ **Comment: Properly answered.**

4. Circularization junctions should be confirmed for each longer GOI (e.g. Spike, Cas9, Factor 8).

Response: We have confirmed the circularization junctions of long GOIs, including EGFP, Fluc, Spike, Cas9, and Factor 8 were by RT-PCR and Sanger sequencing (Extended Data Fig. 3 g-h).

→ **Comment: Properly answered.**

5. In figure 2i, why circRNA bands are much thicker than 0 min precursor RNA bands? Same amounts of RNAs should be loaded to compare precisely.

6. In figure 3g, h, i, why Spike’s nicked circRNA band is much thicker than precursor RNA band? And the RNA amounts in lanes Circ(-) and Circ(+) in each GOI are different. Same amounts of RNAs should be loaded to

compare precisely.

7. In figure 5c, why the bands of precursor RNAs and circRNAs are much thicker than 0 min precursor RNA bands? Same amounts of RNAs should be loaded to compare precisely.

8. In figure 5d, why precursor RNAs are much thicker than 0 min precursor RNA bands? Same amounts of RNAs should be loaded to compare precisely.

Response to Q5-8: Circ (-) lanes in these gels are used to indicate position of precursors in circ (+) lanes, not for intensity comparison.

→ **Comment:** In those figures, 0 min samples or (-) samples should be the “before” reaction samples for intensity comparison at 0 min time point or for reaction (-). I don’t understand why 0 min or (-) samples are added only to indicate position of precursor by adding different amounts. The samples at 0 min or (-) reaction will provide the important information about how much amounts of the precursors were self-circularized or self-cleaved/hydrolysed.

9. TRIC and PIE are based on the same ribozyme, Anabaena group I intron, and use two consecutive transesterification reactions for the splicing. In line 162-165, is there any possible reason why the ribozyme kinetics of TRIC is better than that of PIE? Any previous studies and/or experimental evidences to support the proposed mechanism for better kinetics of TRIC over PIE should be provided.

Response: The function of group I introns require not just the intron (ribozyme) but also exon sequences which form the P1 and P10 structures with the ribozyme. Previous studies have shown that apart from P1 and P10, exon structure -the anticodon stem, in the case of the Anabaena group I intron, are also important for the ribozyme activity (Ref. 25). In TRIC, we have strengthened the anticodon stem which increased the efficiency of TRIC.

→ **Comment:** Would it be fair to compare with PIE without the strengthened anticodon to conclude ribozyme kinetics of TRIC is better than that of PIE? I’d like to suggest you to perform the comparison of kinetics b/w PIE with the strengthened anticodon stem and TRIC.

10. In line 166-167, concatenations seemed to be observed for TRIC V2 also, even though less than PIE. Is this because of post-transcriptional circularization using “high concentrations of precursor RNAs”, as observed for chemical or ligase based method using precursor RNAs? The generation of concatenations would cause inefficient large scale production of circRNA. Then, post-transcriptional circularization proposed in this study would be also impractical for large scale circRNA production, compared to current co-transcriptional method. Concatenations observed for post-transcriptional circularization using TRIC V2 should be more reduced

Response: We used 1 μ M FL for post-transcriptional circularization (~670 ng/ μ l for EGFP), while in IVT, the concentration of RNA is usually >4,000 ng/ μ l. If concatenations are due to ‘high concentrations of precursor RNAs’, co-transcriptional circularization would produce more concatenations. Post-transcriptional circularization has several advantages including higher circular RNA yield due to less co-transcriptional nicking (see point 3), much cleaner product (see Fig. 5b), better controlled buffer condition, lower Mg²⁺ concentration. Particularly, as one of our key conclusions, the FL-post transcriptional circularization protocol is essential for synthesis of circRNAs >5,000 nt (see section A new protocol enables synthesis of circRNAs >8,000 nt).

→ **Comment:** Properly answered.

11. In Figure 5g, what is the light salmon-colored column? Nicking ratio?

Response: Yes, the salmon-coloured bar corresponds to the nicking ratio. A legend has been added for clarity.

→ **Comment:** Properly answered.

12. Is AKTA pure micro system FLPC or HPLC? Chromatograms and purification condition to check circRNA peak among various RNAs and the efficiency during purification should be presented.

Response: For size-exclusion chromatography-based purification of circRNAs we were using a Sepax SRT SEC-2000 HPLC column (cat. number 215980P-4630) operated on a AKTA pure micro FPLC system at 10 MPa pressure limit. The AKTA pure micro system was operated in the absence of UV illumination to reduce nicking during the runs. As a result, we analysed all runs by agarose gel electrophoresis and we cannot provide chromatography profiles. Additionally, each purified circRNAs was quality-checked by urea-agarose gel electrophoresis, as presented in Fig. 6 and Extended Data Fig. 7.

→ **Comment:** In Reference 6 (Nat. Commun 9, 2629 (2018)), the samples were also collected w/o UV illumination to avoid nicking, but chromatograms were shown by using UV detector to provide useful information in Ref 6. I'd like to suggest that you need to provide chromatogram by using UV detector (Nicking will be not problematic in the short time it passes through UV detector and/or some nicking can be considered due to UV illumination), which can be compared with Ref 6's PIE chromatogram and give useful information for readers in terms of purification efficiency, by-product pattern etc.

13. In line 262-264, "Recent studies have reported Tetrahymena thermophila group I intron-derived constructs for clean RNA circularization. However, these constructs resemble the V1.30, exhibiting lower efficiency compared to V1.0 and PIE.

→ The strategy in Reference 40 & 41 utilized different kind of group I intron (Tetrahymena group I intron) from V1.30, so how can you expect it to be less efficient? On what basis are the structures from Tetrahymena group I intron similar to the V1.3 structure? In the Reference 40 and 41, the circularization efficiency was observed to be comparable to or more than PIE. Direct experimental comparison between different constructs should be performed to conclude.

Response: As explained in point 9, both the STS construct (Ref. 29) and the Ribozyme constructs (Ref. 30) contain minimal exon structures and thus are similar to the TRIC-V1.30. We have prepared Extended Data Fig. 5 to explain V1.3 constructs. None of the previous studies has done a head-to-head comparison with PIE in terms of circularization efficiency. Our opinion is that none of those constructs is comparable to PIE. To provide a head-to-head comparison between TRIC-V2 and the above mentioned two constructs, we cloned CVB3-EGFP to the best STS (AU-rich no. 16) and ribozyme (CVB3IRES-GFP, Fig. 6) constructs. As shown in Fig. 5 g-h, TRIC-V2 outperformed both constructs. Importantly, optimized features from Ana intron can be transferred to the Tetra intron.

→ **Comment:** Properly answered.

14. In line 343-345, is it possible that 55 °C 20 min heating increases nicking and degradation of RNAs? The condition with 10 mM MgCl₂ and heating for post-transcriptional method would be favorable condition for nonspecific nicking and hydrolysis.

Response: We agree that extensive circularization will eventually lead to nicking of circRNAs. In practice, the circularization time is dependent on factors such as the size of the FL, the precise architecture of the TRIC, etc. and should thus be adjusted to achieve the best circRNA yield vs nicking ratio for the respective circRNA. In comparison, during IVT with high Mg²⁺, which is performed for hours, co-transcriptional circularization cannot be avoided and causes more severe nicking than controlled post-transcriptional circularization (see above 10 and Fig. 2).

→ **Comment:** Properly answered.

15. In Fig.7, the authors showed higher translation efficacy of TRIC circRNAs, compared to PIE circRNAs. If the expression efficacy of TRIC circRNAs is really higher than PIE circRNAs, what is the reason for higher

expression? Comparison of innate immunity data between TRIC and PIE seems inconclusive. To compare the translation efficacy more precisely, the exact amount of circRNAs present in the transfected cells should be also compared.

Response: While we did not measure intracellular circRNA levels, the transfections were performed identically, and TRIC circRNA showed similar expression with PIE circRNA on day 1 (Fig. 7a), which is also consistent with the similarly day 1 immunogenicity. The expression difference after day 1 could result from less full-length contamination due to higher V2 circularization efficiency or the absence of bacterial sequence and use of section of human 28S rRNA as we have mentioned in the discussion.

→ No comments

16. In the rolling circle translation (RCT) approach, isn't it possible that peptide fragment downstream P2A site of the GOI may be generated via readthrough translation of modified IRES? If so, the unwanted products may cause serious concerns to the cells/tissue administered the circRNAs that can perform RCT. How you can remove the unwanted expression products from RCT circRNA?

Response: Translation of the circCSFV-Nluc_RCT construct will indeed produce Nluc in the first and translated IRES -Nluc fusion proteins in all subsequent translation rounds. We agree that it will be important to characterize the IRES-derived translation products in future studies (see Discussions), and to design or screen for potent, small IRESs that are non-toxic when translated.

→ No comments

17. Several previous studies proposed efficient circRNA generation without any extraneous sequences caused by PIE method. To show more advanced improvements and provide practical implication using the method developed in this study, in vivo expression data showing more durable and higher expression of GOI should be provided and compared.

Response: We agree with the reviewer that it would be interesting to see how TRIC circRNAs behave in vivo and this is certainly the next step. However, in vivo expression experiments are beyond the scope of the current study.

→ No comments

Rebuttal 1

Reviewer #1 (Report for the authors (Required)):

The manuscript presents a novel trans splicing-based approach called Trans Ribozyme-based Circularization (TRIC), for the efficient synthesis of circular RNAs (circRNAs) that are over 8,000 nucleotides (nt) in length. The study also explores the use of viral Internal Ribosomal Entry Sites (IRESs) for Rolling Circle Translation (RCT) and assesses the immunogenicity and translation efficiency of the synthesized circRNAs. The TRIC method for circRNA synthesis is well-described, and the use of the Anabaena tRNA Leu group I intron as a trans ribozyme is justified. Clear figures and gel electrophoresis results convincingly support the efficiency of the TRIC approach. The optimization steps (e.g., introduction of an extended guide sequence and internal loops) are logically presented and supported by experimental data. The comparison of TRIC (V1.0) with the permuted intron-exon method (PIE) for circularizing various genes is informative. The rationale for suppressing co-transcriptional circularization to reduce RNA nicking is well-explained.

The experimental design was thorough, the study was well-executed, and the findings had potential implications for the advancement of mRNA therapeutics. The manuscript is generally well-structured. Thus, I believe the manuscript is suitable for publication pending minor revisions and clarification in certain sections.

Specific Comments:

1. The abstract provides a clear overview of the study, highlighting the main findings and implications. However, it was inappropriate to claim that circRNAs incorporated sequences from human 28S ribosomal RNA have low immunogenicity and are translated more efficiently than PIE circRNAs because PIE is a method to generate the circRNAs. The translation efficiency should be compared between the circular RNA with or without human 28S ribosomal RNA.

Response: We did mean to compare the translation efficiency between circRNAs with or without human 28S ribosomal RNA. We have revised the "PIE circRNAs" to "PIE-derived circRNAs".

2. The abstract of the manuscript also requires clarification regarding the claim of a protein expression increase of more than 7,000-fold by utilizing viral internal ribosomal entry sites (IRESs) for the rolling circle translation (RCT). It should be explicitly stated that this increase is in comparison to a human OR4F17 IRES instead of the single-shot translation. Such a clarification will ensure accurate interpretation of the results and prevent readers from drawing inaccurate conclusions.

Response: The abstract has been revised. The comparison is between the translation efficiency of RCT using a viral IRES and the translation efficiency of previous RCT constructs. We selected a known potent RCT construct (OR4F17, Fan, X., et al., 2022) as a reference and showed an over 7,000-fold of increasing in efficiency. We didn't mean to compare efficiency of RCT with efficiency of single-shot translation either from circRNA or linear mRNA.

3. In page 4 of the manuscript and as depicted in Figure 3, the length of P2A-EGFP is 813 nt, while FL EGFP measures 1963 nt. Please explicitly delineate the different modules of FL EGFP resulted in longer sequence compared to P2A-EGFP in the manuscript.

Response: FL EGFP additionally contains the 741 nt of CVB3 IRES while FL P2A-EGFP does not. Thus, the FL/circular EGFP and FL/circular P2A-EGFP RNAs are 1963/1638 nt and 1142/813 nt, respectively. We have added additional explanations in the manuscript.

4. The exploration of the modified circRNAs using TERIC approach is innovative. The results indicating the synthesis of modified circZnf609 are compelling. However, circZnf609, being an endogenous molecule, inherently lacks immunogenicity. Please discuss the rationale for choosing circZnf609 instead of virus circSpike for testing immunogenicity.

Response: We have tested immunogenicity of two exogenous circRNAs, namely EGFP and Nluc, so for the third candidate we selected an endogenous one, the circZnf609. CircZnf609 is expected to be nonimmunogenic and PRH purified circZnf609 was indeed nonimmunogenic, confirming that PRH is sufficient for high quality circRNA preparation. We didn't expect modified circRNAs to be less immunogenic than PRH purified circRNAs as immunogenicity of PRH purified circRNA was already very low. Immunogenicity test of modified circRNAs confirmed this.

5. The investigation of translation efficiency and the use of viral IRESs for RCT are interesting. The results, especially the significant increase in translation efficiency with CSFV IRES-RCT, are noteworthy. However, in Fig. 7f, the expression level of V2-CVB3-Nluc is higher than those of N1 pseudouridine-modified Nluc and CFSV-Nluc-RCT. It will be very interesting if the author can show the RCT of V2 circCVB3-Nluc also can get more than 10-fold enhancement than V2 circCVB3-Nluc only.

Response: We thank the reviewer for this suggestion. However, engineering of the CVB3 IRES into a RCT is

more challenging than the CSFV IRES due to its larger size and complexity. This will require extensive optimization efforts and is thus currently beyond the scope of the current research.

6. During the RCT, as the authors described that the stop codons have been eliminated, the IRES sequence is also translated, leading to the synthesis of an IRES encoded additional peptide. Does the N-terminus of the POI carry at least a portion or the entire IRES translated peptide sequence after P2A cleavage? This is my main concern, and I hope the authors could provide a Western blot to show the protein molecular weight to confirm it contains the additional IRES peptide or not.

Response: During RCT, the first round of translation would produce the Nluc, and second and subsequent rounds of translation would produce the Nluc fused with N-terminal IRES peptide (IRES-Nluc). Following the reviewer's suggestion, we checked the size of RCT translation products by Western blotting and found that both Nluc and IRES-Nluc were observed. As expected from multiple rounds of translation of the same circRNA, a higher level of IRES-Nluc was produced (Extended Data Fig. 10c). As has mentioned in the discussion, further studies are needed to characterize IRES-derived translation products. Additionally, we could also see a small number of polyproteins, likely due to incomplete 2A cleavage, suggesting that highly efficient cleavage sequences are needed for RCT.

Reviewer #2 (Report for the authors (Required)):

Du et al. presented Anabaena group I intron-based trans ribozyme approach for circular RNA generation (TRIC approach). They showed that the method is independent of bacterial sequences, outperforms PIE method that is currently the most utilized method for RNA circularization, and allows for incorporation of RNA modification. Moreover, they proposed rolling circle translation (RCT)-based enhancement of GOI expression by utilizing and modifying viral IRES.

There are debating concerns in terms of potential innate immunity induction in the bacterial sequences remaining in the circular RNA generated by PIE method. To overcome the issues, several papers recently described already group I intron-based trans ribozyme approach to generate circular RNA without undesired intronic sequences (NAR 51, e78 (2023), Mol Ther Nucleic Acids 33, 587-598 (2023): references 40 and 41 in this manuscript). Main difference of the present study from the previous ones is that the authors here engineered, modified, and optimized the composition of Anabaena tRNA^{Leu} group I intron, in contrast that previous papers engineered Tetrahymena group I intron. Therefore, the concept of the trans ribozyme approach was already suggested by other groups. Moreover, group II-ribozyme based PIE method was also suggested as the circularization approach including the desired GOI only (bioRxiv, 2022, doi: <https://doi.org/10.1101/2022.05.31.494115>). Although the authors proposed the RNA circularization method superior to the current PIE method and efficient RCT method capable of highly improving GOI expression, following concerns and lack of clear data presentation should be addressed.

1. Line 42 and 208-209: "its inability to synthesize modified circRNAs", "The PIE, and likely the TRIC as well, are unable to synthesize modified circRNAs since modifications could disrupt the ribozyme activity.

 That's not true. 5% m6A-modified circRNA was reported to be generated using PIE method (Nat. Biotechnol. 2023, 41, 262-272: Reference 7 in the manuscript). Certain % of modified nucleotide incorporation condition does not affect ribozyme folding and its activity.

Response: We meant 100% modified, as can be seen in paragraph 2 of section "**Synthesis and immunogenicity of modified circRNAs**". To clarify this point, 'fully' has been added before 'modified circRNAs'.

2. Line 45-46: "Urea-polyacrylamide gels effectively distinguish circRNAs from their linear forms but are limited to <500 nt."

 Longer circRNAs (~1000-nt) also were reported to be efficiently distinguished from their linear forms in urea-polyacrylamide gel (Mol Ther Nucleic Acids 2023, 33, 587-598: Reference 41 in the manuscript).

Response: We agree that circRNAs >500 nt may also be separated by urea-PAGE if required. Our rationale for suggesting urea-PAGE is limited to <500 nt (see reference 18, page 225, note 5) is that analysing long circRNAs in urea-PAGE requires low percentages of acrylamide (4% or lower), which increases gel handling difficulties, and long running times. Both of these make analysis of long circRNAs (>500 nt) by urea-PAGE impractical.

3. Line 91-92: if the concentration of Mg²⁺ and NTP were lowered to avoid nicking during IVT, transcription yield would be also lower. How much transcription yield was affected by lower concentrations of Mg²⁺ and NTP? Is it okay for large scale production of circRNA?

Response: We thank the reviewer for making this important point. We measured the total RNA yield under our IVT conditions (6 h) using Mg²⁺ concentrations between 10 and 24 mM and found that the yield remained unchanged when Mg²⁺ is higher than 12 mM (Extended Data Fig. 2f). As a result, using Mg²⁺ concentrations between 12-16 mM would lead to higher circRNA yield compared to 18 – 24 mM, as co-transcriptional circRNA nicking is reduced.

4. Circularization junctions should be confirmed for each longer GOI (e.g. Spike, Cas9, Factor 8).

Response: We have confirmed the circularization junctions of long GOIs, including EGFP, Fluc, Spike, Cas9, and Factor 8 were by RT-PCR and Sanger sequencing (Extended Data Fig. 3 g-h).

5. In figure 2i, why circRNA bands are much thicker than 0 min precursor RNA bands? Same amounts of RNAs should be loaded to compare precisely.

6. In figure 3g, h, i, why Spike's nicked circRNA band is much thicker than precursor RNA band? And the RNA amounts in lanes Circ(-) and Circ(+) in each GOI are different. Same amounts of RNAs should be loaded to compare precisely.

7. In figure 5c, why the bands of precursor RNAs and circRNAs are much thicker than 0 min precursor RNA bands? Same amounts of RNAs should be loaded to compare precisely.

8. In figure 5d, why precursor RNAs are much thicker than 0 min precursor RNA bands? Same amounts of RNAs

should be loaded to compare precisely.

Response to Q5-8: Circ (-) lanes in these gels are used to indicate position of precursors in circ (+) lanes, not for intensity comparison.

9. TRIC and PIE are based on the same ribozyme, Anabaena group I intron, and use two consecutive transesterification reactions for the splicing. In line 162-165, is there any possible reason why the ribozyme kinetics of TRIC is better than that of PIE? Any previous studies and/or experimental evidences to support the proposed mechanism for better kinetics of TRIC over PIE should be provided.

Response: The function of group I introns require not just the intron (ribozyme) but also exon sequences which form the P1 and P10 structures with the ribozyme. Previous studies have shown that apart from P1 and P10, exon structure -the anticodon stem, in the case of the Anabaena group I intron, are also important for the ribozyme activity (Ref. 25). In TRIC, we have strengthened the anticodon stem which increased the efficiency of TRIC.

10. In line 166-167, concatenations seemed to be observed for TRIC V2 also, even though less than PIE. Is this because of post-transcriptional circularization using "high concentrations of precursor RNAs", as observed for chemical or ligase based method using precursor RNAs? The generation of concatenations would cause inefficient large scale production of circRNA. Then, post-transcriptional circularization proposed in this study would be also impractical for large scale circRNA production, compared to current co-transcriptional method. Concatenations observed for post-transcriptional circularization using TRIC V2 should be more reduced

Response: We used 1 μ M FL for post-transcriptional circularization (~670 ng/ μ l for EGFP), while in IVT, the concentration of RNA is usually >4,000 ng/ μ l. If concatenations are due to 'high concentrations of precursor RNAs', co-transcriptional circularization would produce more concatenations. Post-transcriptional circularization has several advantages including higher circular RNA yield due to less co-transcriptional nicking (see point 3), much cleaner product (see Fig. 5b), better controlled buffer condition, lower Mg^{2+} concentration. Particularly, as one of our key conclusions, the FL-post transcriptional circularization protocol is essential for synthesis of circRNAs >5,000 nt (see section **A new protocol enables synthesis of circRNAs >8,000 nt**).

11. In Figure 5g, what is the light salmon-colored column? Nicking ratio?

Response: Yes, the salmon-coloured bar corresponds to the nicking ratio. A legend has been added for clarity.

12. Is AKTA pure micro system FLPC or HPLC? Chromatograms and purification condition to check circRNA peak among various RNAs and the efficiency during purification should be presented.

Response: For size-exclusion chromatography-based purification of circRNAs we were using a Sepax SRT SEC-2000 HPLC column (cat. number 215980P-4630) operated on a AKTA pure micro FPLC system at 10 MPa pressure limit. The AKTA pure micro system was operated in the absence of UV illumination to reduce nicking during the runs. As a result, we analysed all runs by agarose gel electrophoresis and we cannot provide chromatography profiles. Additionally, each purified circRNAs was quality-checked by urea-agarose gel electrophoresis, as presented in Fig. 6 and Extended Data Fig. 7.

13. In line 262-264, "Recent studies have reported Tetrahymena thermophila group I intron-derived constructs for clean RNA circularization. However, these constructs resemble the V1.30, exhibiting lower efficiency compared to V1.0 and PIE.

→ The strategy in Reference 40 & 41 utilized different kind of group I intron (Tetrahymena group I intron) from V1.30, so how can you expect it to be less efficient? On what basis are the structures from Tetrahymena group I intron similar to the V1.3 structure? In the Reference 40 and 41, the circularization efficiency was observed to be comparable to or more than PIE. Direct experimental comparison between different constructs should be performed to conclude.

Response: As explained in point 9, both the STS construct (Ref. 29) and the Ribozyme constructs (Ref. 30) contain minimal exon structures and thus are similar to the TRIC-V1.30. We have prepared Extended Data Fig. 5 to explain V1.3 constructs.

None of the previous studies has done a head-to-head comparison with PIE in terms of circularization efficiency. Our opinion is that none of those constructs is comparable to PIE. To provide a head-to-head comparison between TRIC-V2 and the above mentioned two constructs, we cloned CVB3-EGFP to the best STS (AU-rich no. 16) and ribozyme (CVB3IRES-GFP, Fig. 6) constructs. As shown in Fig. 5 g-h, TRIC-V2 outperformed both constructs. Importantly, optimized features from Ana intron can be transferred to the Tetra intron.

14. In line 343-345, is it possible that 55 °C 20 min heating increases nicking and degradation of RNAs? The

condition with 10 mM MgCl₂ and heating for post-transcriptional method would be favorable condition for nonspecific nicking and hydrolysis.

Response: We agree that extensive circularization will eventually lead to nicking of circRNAs. In practice, the circularization time is dependent on factors such as the size of the FL, the precise architecture of the TRIC, etc. and should thus be adjusted to achieve the best circRNA yield vs nicking ratio for the respective circRNA. In comparison, during IVT with high Mg²⁺, which is performed for hours, co-transcriptional circularization cannot be avoided and causes more severe nicking than controlled post-transcriptional circularization (see above 10 and Fig. 2).

15. In Fig.7, the authors showed higher translation efficacy of TRIC circRNAs, compared to PIE circRNAs. If the expression efficacy of TRIC circRNAs is really higher than PIE circRNAs, what is the reason for higher expression? Comparison of innate immunity data between TRIC and PIE seems inconclusive. To compare the translation efficacy more precisely, the exact amount of circRNAs present in the transfected cells should be also compared.

Response: While we did not measure intracellular circRNA levels, the transfections were performed identically, and TRIC circRNA showed similar expression with PIE circRNA on day 1 (Fig. 7a), which is also consistent with the similarly day 1 immunogenicity. The expression difference after day 1 could result from less full-length contamination due to higher V2 circularization efficiency or the absence of bacterial sequence and use of section of human 28S rRNA as we have mentioned in the discussion.

16. In the rolling circle translation (RCT) approach, isn't it possible that peptide fragment downstream P2A site of the GOI may be generated via readthrough translation of modified IRES? If so, the unwanted products may cause serious concerns to the cells/tissue administered the circRNAs that can perform RCT. How you can remove the unwanted expression products from RCT circRNA?

Response: Translation of the circCSFV-Nluc_RCT construct will indeed produce Nluc in the first and translated IRES -Nluc fusion proteins in all subsequent translation rounds. We agree that it will be important to characterize the IRES-derived translation products in future studies (see Discussions), and to design or screen for potent, small IRESs that are non-toxic when translated.

17. Several previous studies proposed efficient circRNA generation without any extraneous sequences caused by PIE method. To show more advanced improvements and provide practical implication using the method developed in this study, in vivo expression data showing more durable and higher expression of GOI should be provided and compared.

Response: We agree with the reviewer that it would be interesting to see how TRIC circRNAs behave in vivo and this is certainly the next step. However, in vivo expression experiments are beyond the scope of the current study.

Reviewer #3 (Report for the authors (Required)):

Du et al. report a novel strategy for the in vitro production of large circular RNAs based on the action of group I intron trans-splicing. In particular, the Anabaena tRNA LEU group I intron was used and engineered for RNA circularisation. A prototype trans acting ribozyme circularisation (TRIC) system was successively optimised, including ribozyme structure and circularisation conditions, to show significantly better performance than the corresponding circularisation by the PIE method. Importantly, the developed TRIC construct has minimal sequence requirements and the circularised RNA product contains only minimal portions of the native splicing sequence. The technique is demonstrated for the efficient circularisation of RNAs between 1000 and 9000 nt in length. Furthermore, the authors have developed a convincing protocol for the analysis and purification of the circularised product based on denaturing agarose gels. Immunogenicity studies were carried out, including the adaptation of TRIC to circularise modified RNA using a trans excision ribozyme strategy. Finally, the authors demonstrate the efficient generation of RCT constructs using their novel technique, which are translated with high efficiency.

This work represents a technical advance with implications for the field of circRNA synthesis and application in molecular biology and medicine. It is a very timely contribution, as the efficient and precise production of circRNAs is of paramount importance for their use in RNA therapies. In addition, the reported analytical developments are relevant to the broader field of large RNA preparation and analysis.

Specific comments

1. The authors should also consider the TORNADO system in their introduction and discussion (Litke & Jaffrey, Nat Biotechnol. 2019; 37(6): 667-75) It was developed for RNA circularisation in cells, but can also be used for RNA circularisation in vitro.

Response: We thank the reviewer for the suggestion. We understand that there are cell-based circRNA synthesis systems including TORNADO, but our research focuses on in vitro circRNA synthesis. We believe the TORNADO system could work in vitro, too, but we could not find any corresponding publication, where this has been tested.

2. The paper is less clear on terminology in some places. For example, when the term circ3*Flag is used for the first time, it would be helpful to explain that this is the desired circularised product. The same applies to ACA. Figure 2a shows that this is the anticodon arm. However, the abbreviation should also be defined in the text when it is used for the first time. There may be other examples of this that require attention.

Response: We have now explained that circ3*Flag represents circular 3*Flag. The full name of ACA is introduced in line 63.

3. Page 3, line 72/73: The authors say: "Due to the minimal FL, we concluded that the TRIC-V0 construct circularizes the 3*Flag sequence efficiently during IVT." This statement needs to be supported by quantitative data. Several byproducts are seen and by visual inspection the amount of remaining FL is not really minimal.

Response: We have performed IVT of V0 three times and checked the co-transcriptional circularized sample by a 6% urea-PAGE and the remaining relative FL ratio is 3.6 (Extended Data Fig. 2 a-b).

4. Page 3, line 76/77: "introducing a 20 or 23 bp EGS and internal loops to TRIC-V0 (Fig.2f)". There is only one internal loop seen in the structures in Fig. 2f. What is meant is probably "differently structured internal loops"? Please clarify.

Response: We meant the loop length and its surrounding sequence are slightly different. Please see inserts of Fig. 2f.

5. Page 4, line78: "SV1.0 yielded the highest circRNA to intron ratio". What is the ratio of this variant compared to all other variants? Quantitative data should be provided.

Response: Similar to the analysis of V0 (question 3), we have performed IVT of V1.0, 1.1, and 1.3 three times and checked the co-transcriptional circularized sample by a 6% urea-PAGE and quantified the circRNA to intron ratio. These ratios are shown in Extended Data Fig. 2a and 2c, and V1.0 yielded the highest circRNA to intron ratio.

6. Page 4, line 83-84: "Gel analysis revealed that V1.0 has a higher co-transcriptional splicing efficiency than the PIE for EGFP,.....and Cas9 (Fig. 2h)." This is an invalid statement as the gel only shows the disappearance of FL and not the appearance of defined splicing products. Although the consumption of FL is likely to be a result of splicing, it cannot be concluded as stated.

Response: We revised 'revealed' to 'suggested'.

7. Page 5, line 142 – 144: How is V2 derived from V1? A sentence or two should be added to make this clear.

Response: We have defined V2 to be constructs that do not rely on native splicing sequences and added this clarification to the manuscript.

8. References: There are a number of references where there is a lack of information. (e.g. 7, 20, 27, 35, 36, 37).

Response: We thank the reviewer for spotting this oversight. The references have been updated.

9. Figure 2h: The vertical black lines indicating splicing products are barely visible and should be highlighted.

Response: Black lines has been changed to blue and blue arrows are added to highlight splicing products for Spike and Cas9.

10. Figure 4: The structural differences between the variants are not well illustrated. A figure showing which parts of TRIC are varied from V1.30-39 and from V1 to V2 would be desirable. What is the structural difference between V1.30 and V1.31? According to Figure 4b, both have L/R: 3/2, no stem, P1/P10?

Response: We have prepared Extended Data Fig. 5 to illustrate this. Comparing with 1.30, a base pair interaction between 5'EGS and exon near 3' splicing site has been introduced (see Extended Data Fig. 5c).

11. Figure 5g: The illustration lacks the meaning of the orange bar. I learned from the caption that it refers to nicking.

Response: We thank the reviewer for making us aware of this, figure 5 has been updated accordingly.

Reviewer #4 (Report for the authors (Required)):

This manuscript presents a novel trans Anabaena tRNA^{Leu} group I intron mediated methods for synthesis of circRNAs which outperform the more general used permuted intron-exon method (PIE) system. The authors demonstrate that the resulting circRNA has low immunogenicity and able to circularize very large circRNA (up to 8000 nts). It is also demonstrated that if stop codons are removed, the translation can efficiently read through highly structured IRESs, enhancing the translation efficiency by multiple rounds of initiation.

The novelty of the method itself is lowered somewhat in the light of the recent method papers (ref 40 and 41) using tetrahymena self-splicing intron, which also is non-permuted and scarless. The authors make comparison to these and find their method only to be slightly better.

Response: We mentioned in the Discussions section that those Tetra constructs resemble the TRIC V1.30 (Extended Data Fig. 5b). V1.30 is much less efficient than V1.0 and PIE. Meanwhile, V1.0 and PIE are much less efficient than TRIC V2. Thus, we expected TRIC V2 is much better than these Tetra constructs. To test this, we cloned CVB3-EGFP to the best STS (AU-rich no. 16) and ribozyme (CVB3IRES-GFP, Fig. 6) constructs. As shown in Fig. 5 g-h, TRIC-V2 is far superior to both constructs. Importantly, Optimized features from Ana intron can be transferred to the Tetra intron without losing circularization efficiency.

In general, the manuscript is well written and covers a lot of details that are of interest to scientists who want to synthesize and purify circRNA and use it as templates for translation. circRNA has recently attracted a lot of attention and this manuscript will be interesting for a broader audience. However, there are some critical points that must be addressed before publication:

A general problem in the data analysis is that the disappearance of precursor is used as a measure of how efficient the splicing reaction is. But the disappearance of something is a poor measure for the yield because degradation of the RNA is a significant factor. Rather, the amount of circRNA that is produced should be used. In one instance the authors use the circRNA for quantification, but the methods for calculations are very poorly described in the methods section.

Response: Random degradation during circularization applies to all RNA species, including precursor, spliced ribozyme, circRNA, and nicked circRNA. Moreover, the amount of final circRNA that is produced largely depends on purification strategy more than on splicing efficiency. We wanted to estimate splicing efficiency of our constructs, so the intensity of each RNA species was analyzed immediately after circularization on gels.

Fig. 1: Please indicate with arrows the 1. and 2. step attacks in b, c and d as in a)

Response: Figure panels 1b, c, and d have been revised as suggested.

Fig. 2b: It is a bit unclear how much belongs to panel a and b.

Response: We have added an outline to visually separate panels b from a.

Fig. 2f: Why is there so much more linear intron in v. 1.1 and 1.2 when there is not an increase in circRNA and there is no precursor left?

Response: We have loaded the same volume of each IVT product to the gel, so the difference could be attributed to difference in IVT yield of these three constructs.

Fig. 2h: It is not possible to conclude anything regarding splicing efficiency from this gel (As it is done in line 83-84).

Response: We have changed 'revealed' in line 83-84 to 'suggested'.

Section about separation on agarose gel (from line 101). The separation of circRNA from precursor is very dependent on the circularization method, so to conclude that it is possible is too general. E.g. for T4 ligation, where the precursor and circle are of same length, separation may not be possible with these gels. Furthermore, it has been shown that EX-gels from Thermo Fisher separate circRNA and linRNA efficiently and quickly, which should be mentioned in the discussion.

Response: Our conclusion is that in native agarose gels, circRNAs can be separated from their linear counterparts (given an identical length, line 101-119). However, proper gel electrophoresis settings, including agarose concentration and running time need to be selected according to RNA length and the separation is small in general (Extended Data Fig. 4).

We have mentioned the use of EX-gels for analysis of circRNAs in the introduction. As has been shown by R. Chen and colleagues, the EX-gels show batch to batch variation. In addition, urea agarose gels are inexpensive in preparation, which makes urea agarose gels a competitive alternative to EX gels.

Extended data figure 3b-c: What are the extra bands at the other samples (not Spike and Cas9)? Should comment on this.

Response: Corresponding labels have been added to the figure.

Extended data figure 3h: What does "Same as Fig.3g" refer to?

Response: We meant to indicate that this gel is identical to the gel in Fig. 3g. We present this gel here for the convenience of comparison.

Fig. 4: How are the efficiencies in this figure quantified? And how long is each circle? The length of the circle of course needs to be considered when quantifying.

Response: Length of circRNAs range from 101 to 141 nt. Splicing efficiency was roughly estimated using the V1.0 pattern as a reference.

Extended data Fig.4. There is not legend to explain all the symbols. Maybe this should be a Table rather than a figure?

Response: Additional explanations have been added to the figure legend.

Fig. 5: Why use gels in this figure that does not separate the circles from nicked RNA? Would make a lot more sense to run it on gels that separate.

Response: We wanted to compare the splicing efficiency of various constructs. Since both circRNA and the corresponding linear form result from splicing, it is easier to compare when keeping circRNA and nicked RNA unseparated.

Fig. 5e: Would the Michaelis-Menten analysis give the same result if you quantify the amount of circRNA produced rather than the amount of precursor that disappears? It is a rather indirect way of quantifying it.

Response: CircRNA purification procedures can cause additional RNA degradation. Thus, gel analysis immediately after splicing reaction could more faithfully reflect splicing kinetics.

Fig. 5f and g: The annotation of the "Nicked" sample is missing in Fig. 5g. Also, was the gel in Fig. 5f used to make the quantifications in 5g? How was it possible to separate the quantification of only the circRNA or the nicked RNA? These bands are so close that there would be intensity bleeding over. Furthermore, what do the 50% and 20% marks in 5g indicate? And why is the figure cut at the top part, so the V2-EGFP at 8 min cannot be seen?

Response: Fig.5 g has been updated. The separation between circRNA and nicked RNA in Fig. 5f (now Extended Data Fig. 6r) can be seen clearly when zoomed in. Those separations are the basis of splitting band intensities in Image J. Red lines (50% and 20%) indicate reported values of yield and nicking for a circEGFP in Ref. 6

Extended figure 6: If this is RH treatment, should the R and HPLC (X1) be switched around in a?

Response: Yes, thank you for pointing out this error!

Extended figure 7B: Typo: "Ptotocol".

Response: Thank you, and now corrected.

Figure 6h: Would have been nice to see the "none" lanes for TERIC as well.

Response: Circularization of unmodified GOI by TERIC has been tested extensively in Fig. 6g, Extended Data Fig. 8-9. Thus, we did not include a 'none' lane here.

The whole RCT section (and in particular fig. 7f): No comments are made about the CVB3 data in this figure. Also, it is misleading to say that "Translation from RCT is increased by over 7,000-fold" (headline), since it is not related to the linear mRNA or CVB3 data at all. So, the comparison does not make sense. In fact, the data that you compare, and which is 7000-fold increased, is only a little more than 50% compared to the linear mRNA. So, this point is clearly overstated and should be moderated – also in the abstract.

Response: The reason that we included expression data of linear mRNA and CVB3 circRNA is that we wanted to show RCT of CSFV is less efficient. We have clearly mentioned in discussion that: Although RCT efficiency using

CSFV IRES is 52.1% of that from N¹Ψ-modified mRNA, integrating other efficient IRESs like CVB3 and HRV-B3 could enhance RCT efficiency further to many times higher than N¹Ψ-modified linear mRNAs. We wanted to say translation efficiency of RCT is increased by over 7,000-fold. This comparison is between RCT using CSFV IRES and the known potent short IRES (OR4F17). The comparison is not between CSFV-RCT and linear mRNA or CVB3 circRNA, as translation of linear mRNA and CVB3 circRNA is not RCT. Both section title and abstract have been revised as suggested to clarify this.

Fig. 7: How was the Luc data measured. The errors in panel a unrealistically low – is that technical replicates?

Response: Those are biological replicates. We think the small errors can be attributed to the use of adherent cells. The errors are much larger when using suspension cells as shown in Extended 10a.

Extended data Fig. 8: Need more explanation.

Response: Additional explanations have now been added.

Methods: In the first paragraph of “Analysis of circularization efficiency, yield, nicking, and ribozyme kinetics”, the quantification technique is very poorly described. If FL means “Full length precursor”, as stated in the beginning, the “spliced FL” does not make sense. Furthermore, what does “proportionally calculated” mean? This section needs clarity to enable the reader to reproduce the quantification.

Response: We changed ‘spliced FL’ to ‘estimated spliced FL’. Additional explanations have been added.

Rebuttal 2

Reviewer #1 (Report for the authors (Required)):

In this revised manuscript, Yifei Du and coworkers have addressed all my concerns and other reviewers from the prior submission. The manuscript has been improved significantly as well as the incorporation of new comparison with other methods. Of particular relevance to my comments, they have now demonstrated IRES-derived translation products by western blots, this was my principal sticking point with the earlier submission, and I thus now recommend publication in Nature Biomedical Engineering.

Reviewer #2 (Report for the authors (Required)):

Authors properly responded to most of raised issues except several points as follows.

5. In figure 2i, why circRNA bands are much thicker than 0 min precursor RNA bands? Same amounts of RNAs should be loaded to compare precisely.

6. In figure 3g, h, i, why Spike's nicked circRNA band is much thicker than precursor RNA band? And the RNA amounts in lanes Circ(-) and Circ(+) in each GOI are different. Same amounts of RNAs should be loaded to compare precisely.

7. In figure 5c, why the bands of precursor RNAs and circRNAs are much thicker than 0 min precursor RNA bands? Same amounts of RNAs should be loaded to compare precisely.

8. In figure 5d, why precursor RNAs are much thicker than 0 min precursor RNA bands? Same amounts of RNAs should be loaded to compare precisely.

Response to Q5-8: Circ (-) lanes in these gels are used to indicate position of precursors in circ (+) lanes, not for intensity comparison.

Comment: In those figures, 0 min samples or (-) samples should be the "before" reaction samples for intensity comparison at 0 min time point or for reaction (-). I don't understand why 0 min or (-) samples are added only to indicate position of precursor by adding different amounts. The samples at 0 min or (-) reaction will provide the important information about how much amounts of the precursors were self-circularized or self-cleaved/hydrolysed.

Response: The primary reason for loading lower amounts of FL in the 0 min or (-) lanes was to address the significant formation of FL dimers that occurs at higher FL concentrations (due to complementary sequences). To clarify this, we have updated the Materials and Methods section by adding: In some cases, a lower amount of FL RNA was loaded onto each gel to minimize FL dimerization.

Moreover, our calculations were primarily based on the intensity of each RNA species in the respective lanes, and therefore, the 0 min or (-) lanes were not included in these calculations. We acknowledge that the 0 min or (-) samples could potentially inform the presence of self-circularization or self-cleavage/hydrolysis. However, we believe that these processes would result in the appearance of new bands, which were not observed in the (-) lanes.

9. TRIC and PIE are based on the same ribozyme, Anabaena group I intron, and use two consecutive transesterification reactions for the splicing. In line 162-165, is there any possible reason why the ribozyme kinetics of TRIC is better than that of PIE? Any previous studies and/or experimental evidences to support the proposed mechanism for better kinetics of TRIC over PIE should be provided.

Response: The function of group I introns require not just the intron (ribozyme) but also exon sequences which form the P1 and P10 structures with the ribozyme. Previous studies have shown that apart from P1 and P10, exon structure -the anticodon stem, in the case of the Anabaena group I intron, are also important for the ribozyme activity (Ref. 25). In TRIC, we have strengthened the anticodon stem which increased the efficiency of TRIC.

Comment: Would it be fair to compare with PIE without the strengthened anticodon to conclude ribozyme kinetics of TRIC is better than that of PIE? I'd like to suggest you to perform the comparison of kinetics b/w PIE with the strengthened anticodon stem and TRIC.

Response: The key difference between TRIC and PIE is that TRIC retains an intact group I intron at the 5' end, facilitating easier ribozyme folding compared to the PIE construct, which splits the ribozyme into two halves. Moreover, optimal ribozyme activity depends on both the group I intron and exon sequences. Our improvements were primarily achieved through optimizations in the exon regions. To avoid confusion, we have revised the term 'ribozyme kinetics' to 'construct kinetics'.

12. Is AKTA pure micro system FLPC or HPLC? Chromatograms and purification condition to check circRNA peak among various RNAs and the efficiency during purification should be presented.

Response: For size-exclusion chromatography-based purification of circRNAs we were using a Sepax SRT SEC-2000 HPLC column (cat. number 215980P-4630) operated on a AKTA pure micro FPLC system at 10 MPa pressure limit. The AKTA pure micro system was operated in the absence of UV illumination to reduce nicking during the runs. As a result, we analysed all runs by agarose gel electrophoresis and we cannot provide chromatography profiles. Additionally, each purified circRNAs was quality-checked by urea-agarose gel electrophoresis, as presented in Fig. 6 and Extended Data Fig. 7.

Comment: In Reference 6 (Nat. Commun 9, 2629 (2018)), the samples were also collected w/o UV illumination to avoid nicking, but chromatograms were shown by using UV detector to provide useful information in Ref 6. I'd like to suggest that you need to provide chromatogram by using UV detector (Nicking will be not problematic in the short time it passes through UV detector and/or some nicking can be considered due to UV illumination), which can be compared with Ref 6's PIE chromatogram and give useful information for readers in terms of purification efficiency, by-product pattern etc.

Response: We thank the reviewer for this suggestion. In response, we have now included a representative profile for the purification of circNluc using the TRIC construct and the PRH protocol. Although size exclusion chromatography under the PRH protocol can yield high-quality circRNA, achieving complete separation between circRNA and residual FL remains challenging due to the minimal size difference between the two.

Reviewer #3 (Report for the authors (Required)):

The authors have adequately addressed the reviewers' comments and made the appropriate improvements. Suggested additional experiments were performed, missing data were added, and the manuscript was revised to correct typographical errors, add missing details, and include new figures. The overall quality and clarity of the manuscript has improved. My personal comments are well considered.

However, there is still one minor technical concern that I would like to mention:

The authors use the disappearance of the FL as an indicator of circularization. In the text, line 73/74, they still say: "Due to the minimal remaining FL, we concluded that the TRIC-V0 construct circularizes the 3*Flag sequence efficiently during IVT (Extended Data Fig. 2 a-b)." Based on the data in the extended data Fig. 2a, the relative FL ratio was calculated to be 3.6. I don't think one can say EFFICIENT circularization just based on the disappearance of FL. The authors should find a way to calculate or at least estimate the amount of circularized product (seen in the gel in Fig. 2a) relative to FL and the other splicing products to have a better measure of circularization efficiency.

Response: We thank the reviewer for this suggestion. We have now used the relative circRNA ratio to estimate the efficiency. The relative circRNA ratio is calculated by dividing the circRNA yield limit by the ratio of circRNA intensity to total intensity. The yield limit is defined as the ratio of the molecular weight of circRNA to that of FL.